# Multiple bumps can enhance robustness to noise in continuous attractor networks

**Raymond Wang[1,2], Louis Kang [2]***

**1** Redwood Center for Theoretical Neuroscience, University of California, Berkeley, Berkeley, California, United States of America, **2** Neural Circuits and Computations Unit, RIKEN Center for Brain Science, Wako, Saitama, Japan

* louis.kang@riken.jp

**Data Availability Statement:** Simulation and analysis code is available at https://github.com/louiskang-group/wang-2022.

**Funding:** RW and LK were funded by RIKEN Center for Brain Science. The funders had no role in study

## Abstract

A central function of continuous attractor networks is encoding coordinates and accurately updating their values through path integration. To do so, these networks produce localized bumps of activity that move coherently in response to velocity inputs. In the brain, continuous attractors are believed to underlie grid cells and head direction cells, which maintain periodic representations of position and orientation, respectively. These representations can be achieved with any number of activity bumps, and the consequences of having more or fewer bumps are unclear. We address this knowledge gap by constructing 1D ring attractor networks with different bump numbers and characterizing their responses to three types of noise: fluctuating inputs, spiking noise, and deviations in connectivity away from ideal attractor configurations. Across all three types, networks with more bumps experience less noise-driven deviations in bump motion. This translates to more robust encodings of linear coordinates, like position, assuming that each neuron represents a fixed length no matter the bump number. Alternatively, we consider encoding a circular coordinate, like orientation, such that the network distance between adjacent bumps always maps onto 360 degrees. Under this mapping, bump number does not significantly affect the amount of error in the coordinate readout. Our simulation results are intuitively explained and quantitatively matched by a unified theory for path integration and noise in multi-bump networks. Thus, to suppress the effects of biologically relevant noise, continuous attractor networks can employ more bumps when encoding linear coordinates; this advantage disappears when encoding circular coordinates. Our findings provide motivation for multiple bumps in the mammalian grid network.

## Author summary

Our brains maintain an internal sense of location and direction so we can, for example, find our way to the door if the lights go off. A class of neural circuits called continuous attractor networks is believed to be responsible for this ability. These circuits must be resilient against the myriad forms of imperfections and random fluctuations present in the brain, which can degrade the accuracy of their encoded information. We have

design, data collection and analysis, decision to publish, or preparation of the manuscript.

**Competing interests:** The authors have declared that no competing interests exist.

discovered a new way in which continuous attractor networks can improve their robustness to noise: they should distribute their activity among multiple regions in the network, called bumps, instead of concentrating it in a single bump. Bump number is a fundamental feature of continuous attractor networks, but its connection to error suppression has never been appreciated. A recent experiment in rodents suggests that one such network indeed contains multiple regions of activity; our finding provides motivation for why such a configuration may have been evolved.

## Introduction

Continuous attractor networks (CANs) sustain a set of activity patterns that can be smoothly morphed from one to another along a low-dimensional manifold [1–3]. Network activity is typically localized into attractor bumps, whose positions along the manifold can represent the value of a continuous variable. These positions can be set by external stimuli, and their persistence serves as a memory of the stimulus value. Certain CAN architectures are also capable of a feature called path integration. Instead of receiving the stimulus value directly, the network receives its changes and integrates over them by synchronously moving the attractor bump [4–6]. Path integration allows systems to estimate an external state based on internally perceived changes, which is useful in the absence of ground truth.

Path-integrating CANs have been proposed as a mechanism through which brains encode various physical coordinates. Head direction cells in mammals and compass neurons in insects encode spatial orientation by preferentially firing when the animal faces a particular direction relative to landmarks (Fig 1A, top; Refs [7] and [8]). They achieve this as members of 1D CANs whose attractor manifolds have ring topologies [9, 10]. For the case of compass neurons,

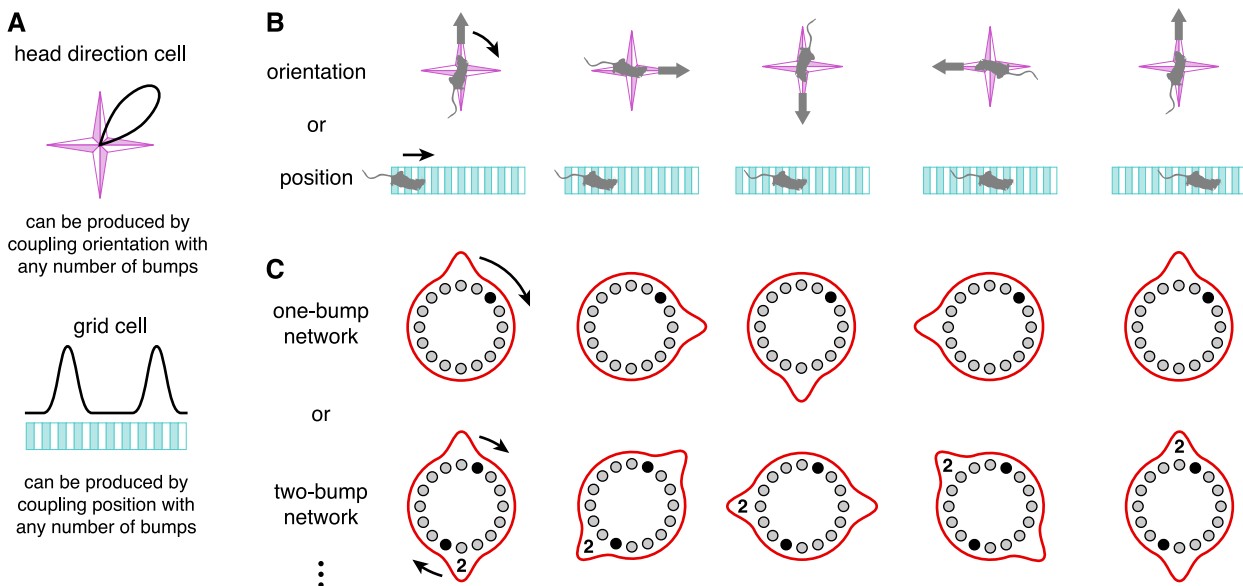

**Fig 1. Continuous attractor networks with any number of bumps can produce head direction cells and grid cells.** (**A**) Desired tuning curves of a head direction cell and a 1D grid cell. (**B**) Orientation and position coordinates whose changes drive bump motion. (**C**) One- and two-bump ring attractor networks. Each black neuron produces the desired tuning curves in **A**. In the two-bump network, the coupling to coordinate changes is half as strong, and the second bump is labeled for clarity.

a ring structure also exists anatomically, and its demonstration of continuous attractor dynamics is well-established [8, 11–13]. Grid cells in mammals encode position by preferentially firing at locations that form a triangular lattice in 2D space (1D analogue in Fig 1A, bottom; Ref [14]). They are thought to form a 2D CAN with toroidal topology [15–18], and mounting experimental evidence supports this theory [19–22]. The ability for head direction cells, compass neurons, and grid cells to maintain their tunings in darkness without external cues demonstrates that these CANs can path integrate [8, 14, 23].

CANs also appear in studies of other brain regions and neural populations. Signatures of continuous attractor dynamics have been detected in the prefrontal cortex during spatial working memory tasks [24–26]. Theorists have further invoked CANs to explain place cells [27, 28], hippocampal view cells [29], eye tracking [4, 6], visual orientation tuning [30, 31], and perceptual decision making [32, 33]. Thus, CANs are a crucial circuit motif throughout the brain, and better understanding their performance would provide meaningful insights into neural computation.

One factor that strongly affects the performance of CANs in path integration is biological noise. To accurately represent physical coordinates, attractor bumps must move in precise synchrony with the animal's trajectory. Hence, the bump velocity must remain proportional to the driving input that represents coordinate changes [18]. Different sources of noise produce different types of deviations from this exact relationship, all of which lead to path integration errors. While noisy path-integrating CANs have been previously studied [10, 18, 34, 35], these works did not investigate of role of bump number. CANs with different connectivities can produce different numbers of attractor bumps, which are equally spaced throughout the network and perform path integration by moving in unison [16, 18, 36]. Two networks with different bump numbers have the same representational capability (Fig 1). They can share the same attractor manifold and produce neurons with identical tuning curves, as long as the coupling strength between bump motion and driving input scales appropriately. The computational advantages of having more or fewer bumps are unknown.

Our aim is to elucidate the relationship between bump number and robustness to noise. We first develop a rigorous theoretical framework for studying 1D CANs that path integrate and contain multiple bumps. Our theory predicts the number, shape, and speed of bumps. We then introduce three forms of noise. The first is Gaussian noise added to the total synaptic input, which can represent fluctuations in a broad range of cellular processes occurring at short timescales. The second is Poisson spiking noise. The third is noise in synaptic connectivity strengths; the ability for bumps to respond readily to driving inputs is generally conferred by a precise network architecture. We add Gaussian noise to the ideal connectivity and evaluate path integration in this setting. The first two forms of noise are independent over time and neurons, in contrast to the third. We find that networks with more bumps can better resist all three forms of noise under certain encoding assumptions. These observations are explained by our theoretical framework with simple scaling arguments. The following Results section presents all simulation findings and major theoretical conclusions; complete theoretical derivations are found in the Theoretical model section.

## Results

### Bump formation in a ring attractor network

We study a 1D ring attractor network that extends the model of Ref [37] to allow for multiple attractor bumps. It contains two neural populations $\alpha \in \{L, R\}$ at each network position $x$, with $N$ neurons in each population (Fig 2A). Each neuron is described by its total synaptic

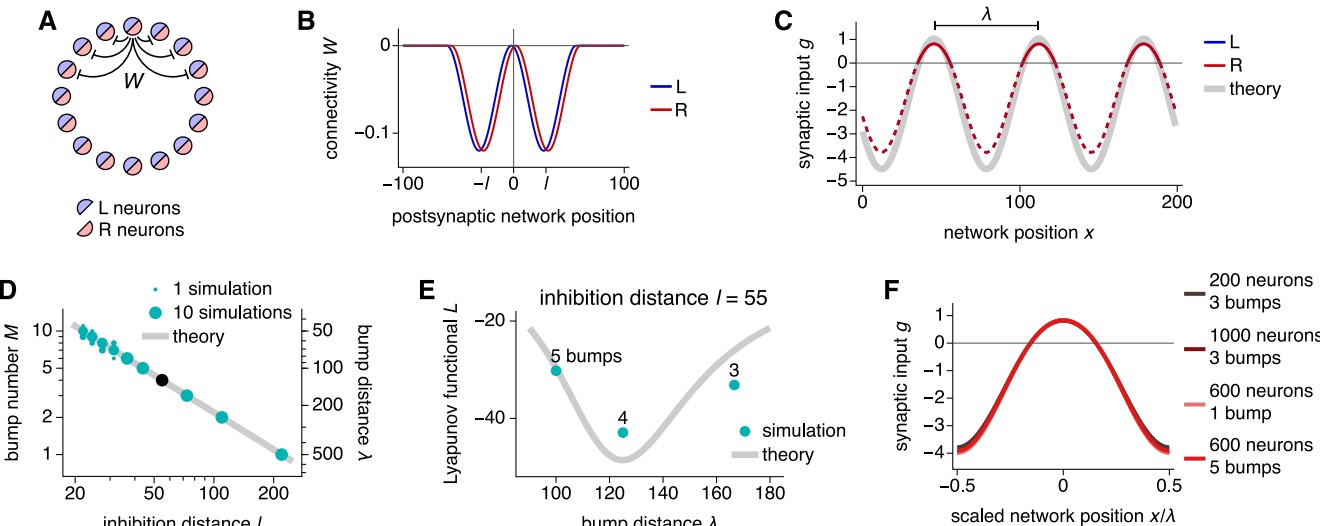

**Fig 2. Bump formation in a ring attractor network.** (**A**) Network schematic with populations L and R and locally inhibitory connectivity W. (**B** and **C**) Networks with 200 neurons and 3 bumps. (**B**) Connectivity weights for a neuron at the origin. The inhibition distance is $l = 29$ and the connectivity shift is $\xi = 2$. (**C**) Steady-state synaptic inputs. Curves for both populations lie over each other. With a ReLU activation function, the firing rates follow the solid portions of the colored lines and are 0 over the dashed portions. The bump distance is $\lambda = 200/3$. Thick gray line indicates Eq 4. (**D** and **E**) Networks with 500 neurons. (**D**) More bumps and shorter bump distances are produced by smaller inhibition distances. Points indicate data from 10 replicate simulations. Line indicates Eq 5. (**E**) The inhibition distance $l = 55$ corresponds to the black point in **D** with $\lambda = 125$ and $M = 4$. These values also minimize the Lyapunov functional (Eq 6), which varies smoothly across $\lambda$ for infinite networks (line) and takes discrete values for finite networks (points). (**F**) The scaled bump shape remains invariant across network sizes and bump numbers, accomplished by rescaling connectivity strengths according to Eq 7. Curves for different parameters lie over one another.

input $g$ that obeys the following dynamics:

$$\tau \frac{dg_\alpha(x,t)}{dt} + g_\alpha(x,t) = \sum_\beta \int dy\, W_\beta(x,y)s_\beta(y,t) + A \pm_\alpha \gamma b(t) + \zeta_\alpha(x,t), \qquad (1)$$

where $\pm_L$ means $-$ and $\pm_R$ means $+$. Aside from spiking simulations, firing rates $s$ are given by

$$s_\alpha(x,t) = \phi[g_\alpha(x,t)], \qquad (2)$$

where $\phi$ is a nonlinear activation function. For all simulations in this Results section, we take $\phi$ to be the rectified linear unit (ReLU) activation function (Eq 35). Our theoretical formulas for diffusion coefficients and velocities in this section also assume a ReLU $\phi$. In S1 Text, we consider a logistic $\phi$ instead and find that all major conclusions are preserved (Fig A in S1 Text), and in the Theoretical methods section, we derive most expressions for general $\phi$. $W$ is the synaptic connectivity and only depends on the presynaptic population $\beta$. It obeys a standard continuous attractor architecture based on local inhibition that is strongest at an inhibition distance $l$. Each population has its synaptic outputs shifted by a small distance $\xi \ll l$ in opposite directions. We use the connectivity profile described in Fig 2B and Eq 38 for all simulations, but all theoretical expressions in this Results section are valid for any $W$. $A$ is the resting input to all neurons. The driving input, or drive, $b$ is proportional to changes in the coordinate encoded by the network; for the physical coordinates in Fig 1B, it represents the animal's velocity obtained from self-motion cues. In our results, $b$ is constant in time. It is coupled to the network with strength $\gamma$. We will consider various forms of noise $\zeta$. Finally, $\tau$ is the neural time constant.

 

With no drive $b = 0$ and no noise $\zeta = 0$, the network dynamics in Eqs 1 and 2 can be simplified to

$$\tau \frac{\mathrm{d}g(x, t)}{\mathrm{d}t} + g(x, t) = 2 \int \mathrm{d}y \, W(x - y)\phi[g(y, t)] + A, \tag{3}$$

where $2W(x - y) = \Sigma_\beta \, W_\beta(x, y)$ and the synaptic inputs $g$ are equal between the two populations. This baseline equation evolves towards a periodic steady-state $g$ with approximate form (see also Ref [38]).

$$g(x) = a \, \cos\frac{2\pi(x - x_0)}{\lambda} + d. \tag{4}$$

Expressions for $a$ and $d$ are given in the Theoretical model section (Eq 60). The firing rates $s(x) = \phi[g(x)]$ exhibit attractor bumps with periodicity $\lambda$, a free parameter that we call the bump distance (Fig 2C). $x_0$ is the arbitrary position of one of the bumps. It parameterizes the attractor manifold with each value corresponding to a different attractor state up to $\lambda$.

The bump number $M = N/\lambda$ is determined through $\lambda$. It can be predicted by the fastest-growing mode in a linearized version of the dynamics (Eq 43; Refs [39] and [40]). The mode with wavenumber $q$ and corresponding wavelength $2\pi/q$ grows at rate $(2\tilde{W}(q) - 1)/\tau$, where $\tilde{W}(q)$ is the Fourier transform of $W(x)$. Thus,

$$\frac{2\pi}{\lambda} = \underset{q}{\mathrm{argmax}}\, \tilde{W}(q). \tag{5}$$

Fig 2D shows that simulations follow the predicted $\lambda$ and $M$ over various inhibition distances $l$. Occasionally for small $l$, a different mode with a slightly different wavelength will grow quickly enough to dominate the network. A periodic network enforces an integer bump number, which discretizes the allowable wavelengths and prevents changes in $\lambda$ and $M$ once they are established. In an aperiodic or infinite system, the wavelength can smoothly vary from an initial value to a preferred length over the course of a simulation [18, 41]. To determine this preferred $\lambda$ theoretically, we notice that the nonlinear dynamics in Eq 3 obey the Lyapunov functional

$$L = - \iint \mathrm{d}x \, \mathrm{d}y \, W(x - y)s(x)s(y) + \int \mathrm{d}x \int_0^{s(x)} \mathrm{d}\rho \, \phi^{-1}[\rho] - A \int \mathrm{d}x \, s(x). \tag{6}$$

In the Theoretical model section, we find for ReLU $\phi$ that $L$ is minimized when $q = 2\pi/\lambda$ maximizes $\tilde{W}(q)$ (Eq 66). This is the same condition as for the fastest-growing mode in Eq 5 (Fig 2E). In other words, the wavelength $\lambda$ most likely to be established in a periodic network is the preferred bump distance in an aperiodic or infinite system, up to a difference of one fewer or extra bump due to discretization.

We now understand how to produce different bump numbers $M$ in networks of different sizes $N$ by adjusting the inhibition distance $l$. To compare networks across different values of $M$ and $N$, we scale the connectivity strength $W$ according to

$$W_\beta(x, y) \propto \frac{M}{N}. \tag{7}$$

This keeps the total connectivity strength per neuron $\int \mathrm{d}x \, W_\beta(x, y)$ constant over $M$ and $N$. In doing so, the shape of each attractor bump as a function of scaled network position $x/\lambda$ remains invariant (Fig 2F). Thus, Eq 7 removes additional variations in bump shape and helps to isolate our comparisons across $M$ and $N$ to those variables themselves. In S1 Text, we

consider the alternative without this scaling and find that many major results are preserved (Fig B in S1 Text).

## Bump dynamics: Path integration and diffusion

The drive $b$ produces coherent bump motion by creating an imbalance between the two neural populations. A positive $b$ increases input to the R population and decreases input to the L population (Fig 3A). Because the synaptic outputs of the former are shifted to the right, the bump moves in that direction. Similarly, a negative $b$ produces leftward bump motion. The bump velocity $v_{\text{drive}}$ can be calculated in terms of the baseline firing rates $s(x)$ obtained without drive and noise (see also Refs [37] and [42]):

$$v_{\text{drive}} = -\frac{\gamma b \xi \int \mathrm{d}x \, \frac{\mathrm{d}^2 s}{\mathrm{d}x^2}}{\tau \int \mathrm{d}x \left(\frac{\mathrm{d}s}{\mathrm{d}x}\right)^2}. \tag{8}$$

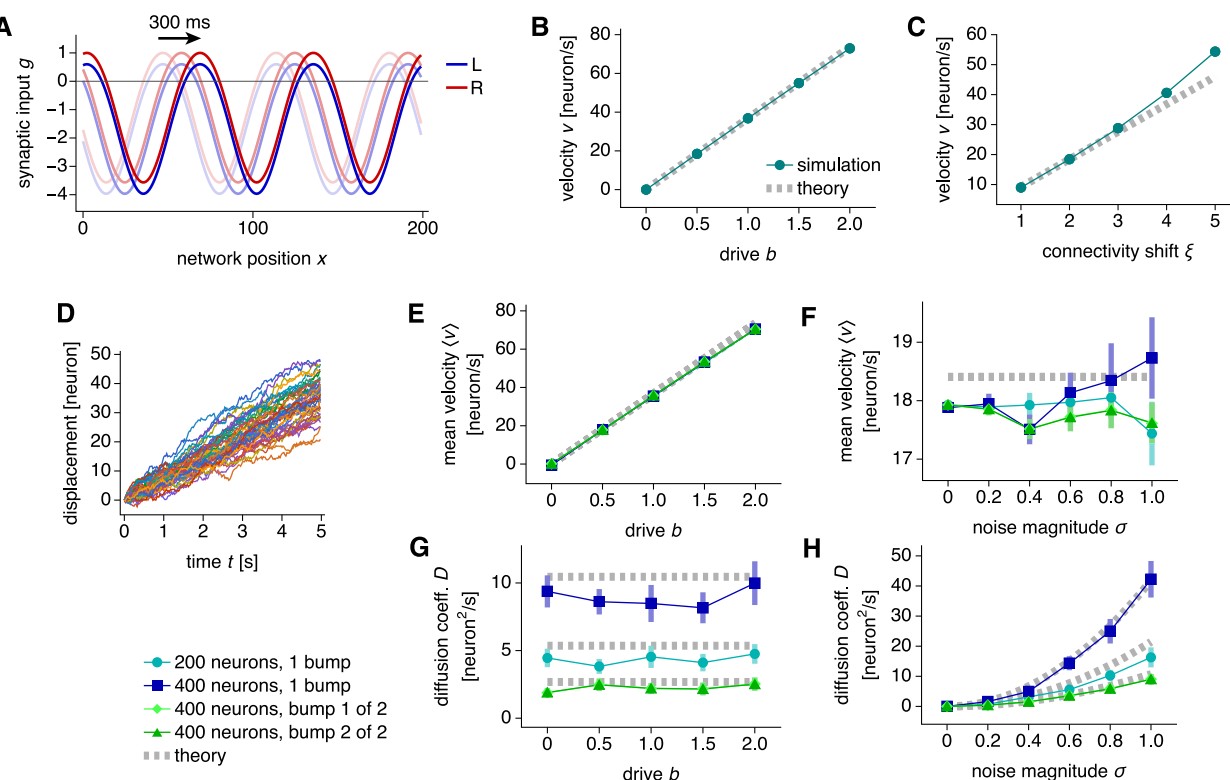

**Fig 3. Dynamics in a ring attractor network.** (**A–C**) Networks with 200 neurons and 3 bumps. (**A**) Synaptic inputs for populations L and R under drive $b = 2$. Snapshots taken at 150 ms intervals demonstrate rightward motion. (**B**) Bump velocity is proportional to drive. The connectivity shift is $\xi = 2$. (**C**) Bump velocity is largely proportional to connectivity shift. The drive is $b = 0.5$. (**D–H**) Networks with synaptic input noise. (**D**) Bump displacements for 48 replicate simulations demonstrating diffusion with respect to coherent motion. Networks with 200 neurons and 1 bump. (**E** and **F**) Mean bump velocity is proportional to drive and remains largely independent of network size, bump number, and noise magnitude. (**G** and **H**) Bump diffusion coefficient scales quadratically with noise magnitude, remains largely independent of drive, and varies with network size and bump number. The noise magnitude is $\sigma = 0.5$ in **D**, **E**, and **G**, and the drive is $b = 0.5$ in **D**, **F**, and **H**. Values for both bumps in two-bump networks lie over each other. Points indicate data from 48 replicate simulations and bars indicate bootstrapped standard deviations. Dotted gray lines indicate Eqs 8 and 10.

As a note, these integrals, as well as subsequent ones, do not include the singular points at the edges of attractor bumps. Eq 8 states that bump velocity is proportional to drive $b$ and connectivity shift $\xi$, which is reflected in our simulations, with some deviation at larger $\xi$ (Fig 3B and 3C). The strict proportionality between $v$ and $b$ is crucial because it implies faithful path integration [18]. If $b(t)$ represents coordinate changes (such as angular or linear velocity in Fig 1B), then the bump position $\theta(t)$ will accurately track the coordinate itself (orientation or position).

In contrast to drive, uncorrelated noise $\zeta$ produces bump diffusion. To illustrate this effect, we introduce one form of $\zeta$ that we call synaptic input noise. Suppose $\zeta$ is independently sampled for each neuron at each simulation timestep from a Gaussian distribution with mean 0 and variance $\sigma^2$. Loosely, it can arise from applying the central limit theorem to the multitude of noisy synaptic and cellular processes occurring at short timescales. Then,

$$\langle \zeta_\alpha(x, t) \rangle = 0, \qquad \langle \zeta_\alpha(x, t) \zeta_\beta(y, t') \rangle = \sigma^2 \Delta t \, \delta(t - t') \delta_{\alpha\beta} \delta(x - y), \tag{9}$$

where the timestep $\Delta t$ sets the resampling rate of $\zeta$, and angle brackets indicate averaging over an ensemble of replicate simulations. Input noise causes bumps to diffuse away from the coherent driven motion (Fig 3D). The mean velocity $\langle v \rangle$ remains proportional to drive $b$, which means that the network still path integrates on average (Fig 3E). Since $\langle v \rangle$ is largely independent of noise magnitude $\sigma$, and the bump diffusion coefficient $D$ is largely independent of $b$, drive and input noise do not significantly interact within the explored parameter range (Fig 3F and 3G). $D$ can be calculated in terms of the baseline firing rates (see also Refs [35] and [43]):

$$D_{\text{input}} = \frac{\sigma^2 \Delta t}{4\tau^2 \int dx \left(\frac{ds}{dx}\right)^2}. \tag{10}$$

The quadratic dependence of $D$ on $\sigma$ is confirmed by simulation (Fig 3H).

We now turn our attention to bump number $M$ and network size $N$. The mean bump velocity $\langle v \rangle$ is independent of these parameters (Fig 3E and 3F), which can be understood theoretically. Bump shapes across $M$ and $N$ are simple rescalings of one another (Fig 2F), so derivatives of $s$ with respect to $x$ are simply proportional to $M$ (more bumps imply faster changes) and inversely proportional to $N$ (larger networks imply slower changes). Similarly, integrals of expressions containing $s$ over $x$ are simply proportional to $N$. In summary,

$$\frac{ds}{dx} \propto \frac{M}{N}, \qquad \frac{d^2s}{dx^2} \propto \frac{M^2}{N^2}, \qquad \int dx \propto N. \tag{11}$$

Applying these scalings to Eq 8, we indeed expect $v_{\text{drive}}$ to be independent of $M$ and $N$. In contrast, Fig 3G and 3H reveal that the diffusion coefficient $D$ varies with these parameters. When a one-bump network is increased in size from 200 to 400 neurons, $D$ increases as well, which implies greater path integration errors. This undesired effect can be counteracted by increasing the bump number from 1 to 2, which lowers $D$ below that of the one-bump network with 200 neurons. These initial results suggest that bump number and network size are important factors in determining a CAN's resilience to noise. We will explore this idea in greater detail.

## Mapping network coordinates onto physical coordinates

Before further comparing networks with different bump numbers $M$ and sizes $N$, we should scrutinize the relationship between bump motion and the physical coordinate encoded by the network. After all, the latter is typically more important in biological settings. First, we consider the trivial case in which each neuron represents a fixed physical interval across all $M$ and $N$; this is equivalent to using network coordinates without a physical mapping (Fig 4A). It is suited for encoding linear variables like position that lack intrinsic periodicity, so larger networks can encode wider coordinate ranges. However, with more bumps or fewer neurons, the range over which the network can uniquely encode different coordinates is shortened. We assume that ambiguity among coordinates encoded by each bump can be resolved by additional cues, such as local features, that identify the true value among the possibilities [44–46]; this process will be examined in detail below. We leave quantities with dimensions of network distance in natural units of neurons.

Multi-bump networks are intrinsically periodic, especially those with a ring architecture. A natural way for them to encode a circular coordinate like orientation would be to match network and physical periodicities. For example, the bump distance may always represent 360˚ across different $M$ and $N$ so that neurons always exhibit unimodal tuning (Fig 4B). This relationship implies that quantities with dimensions of network distance should be multiplied by powers of the conversion factor

$$\frac{360° \cdot M}{N}, \tag{12}$$

which converts units of neurons to degrees. In other words, circular mapping implies normalizing network distances by the bump distance $\lambda = N/M$.

For circular mapping, we must also ensure that networks with different bump numbers $M$ and sizes $N$ path integrate consistently with one another. The same drive $b$ should produce the same bump velocity $v$ in units of degree/s. To do so, we rescale the coupling strength $\gamma$ only under circular mapping:

$$\gamma \propto \frac{N}{M}. \tag{13}$$

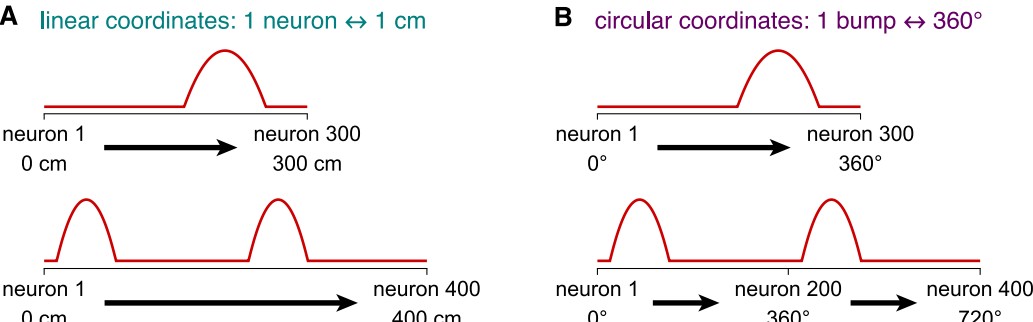

**Fig 4. Possible mappings between network coordinates and two types of physical coordinates. (A)** In networks encoding linear coordinates such as position, one neuron always represents a fixed physical interval. This mapping is trivial and identical to using network coordinates. (**B**) In networks encoding circular coordinates such as orientation, the bump distance always represents 360˚.

This effectively compensates for the factor of $M/N$ in Eq 12. To see this explicitly, recall that $v_{\mathrm{drive}}$ does not depend on $M$ and $N$ in units of neuron/s, as shown in Fig 3E and 3F and previously explained through scaling arguments. Under circular mapping, $v_{\mathrm{drive}}$ would be multiplied by one power of the conversion factor in Eq 12. Since its formula contains $\gamma$ in the numerator (Eq 8), $v_{\mathrm{drive}}$ receives an additional power of the rescaling factor in Eq 13. The two factors cancel each other, so $v_{\mathrm{drive}}$ does not depend on $M$ and $N$ under either mapping:

$$v_{\mathrm{drive}} \propto 1 \quad \text{linear}, \qquad v_{\mathrm{drive}} \propto 1 \quad \text{circular}. \tag{14}$$

Thus, a consistent relationship between $b$ and $v_{\mathrm{drive}}$ is preserved in units of both neurons/s and degrees/s.

Of course, there are other possible mappings between network and physical coordinates across bump numbers and network sizes. For example, intermediate scalings can be achieved with the conversion factor $(M/N)^\mu$ for $0 < \mu < 1$ instead of Eq 12, with the corresponding $\gamma \propto (N/M)^\mu$ instead of Eq 13. But for the rest of our paper, we will consider the linear and circular cases, which correspond to $\mu = 0$ and $\mu = 1$, respectively. To be clear, networks with the same ring architecture are used for both mappings. We will see how noise affects encoding quality in either case.

## More bumps improve robustness to input and spiking noise under linear mapping

We now revisit the effect of input noise on bump diffusion, as initially explored in Fig 3D–3H. We measure how the diffusion coefficient $D$ varies with bump number $M$ and network size $N$ under linear and circular mappings. Under linear mapping, $D$ decreases as a function of $M$ but increases as a function of $N$ (Fig 5A and 5B). Thus, more bumps attenuate diffusion produced by input noise, which is especially prominent in large networks. However, for circular coordinates, $D$ remains largely constant with respect to $M$ and decreases with respect to $N$ (Fig 5A and 5B). Increasing the number of bumps provides no benefit. These results can be understood through Eqs 10, 11 and 12, which predict

$$D_{\mathrm{input}} \propto \frac{N}{M^2} \quad \text{linear}, \qquad D_{\mathrm{input}} \propto \frac{1}{N} \quad \text{circular}. \tag{15}$$

Two powers of the conversion factor in Eq 12 account for the differences between the two mappings.

Next, we investigate networks with spiking noise instead of input noise. To do so, we replace the deterministic formula for firing rate in Eq 2 with

$$s_\alpha(x, t) = \frac{c_\alpha(x, t)}{\Delta t}. \tag{16}$$

Here, $s$ is a stochastic, instantaneous firing rate given by the number of spikes $c$ emitted in a simulation timestep divided by the timestep duration $\Delta t$. We take the $c$'s to be independent Poisson random variables driven by the deterministic firing rate:

$$c_\alpha(x, t) \sim \mathrm{Pois}[\phi[g_\alpha(x, t)]\Delta t]. \tag{17}$$

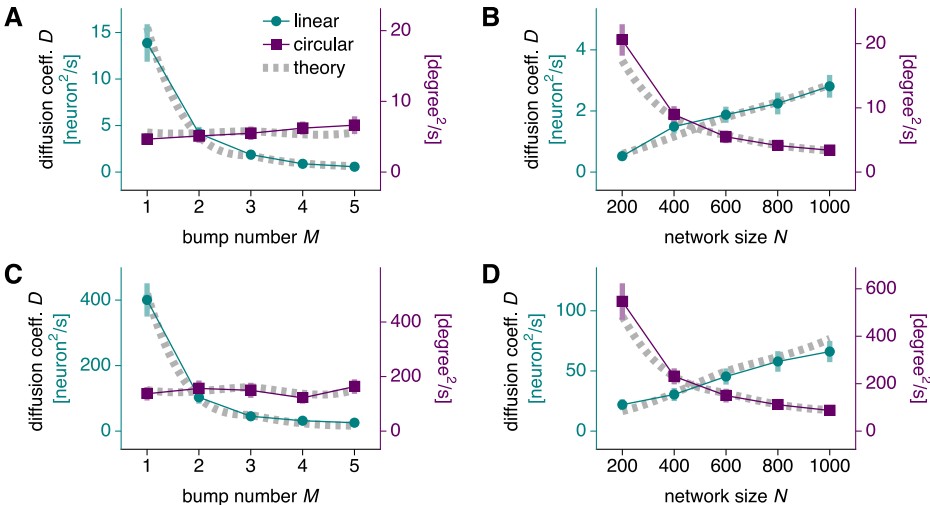

**Fig 5. Bump diffusion due to input and spiking noise.** (A, B) Networks with synaptic input noise of magnitude $\sigma =$ 0.5 and drive $b = 0.5$. Dotted gray lines indicate Eq 10. (**A**) Diffusion decreases with bump number under linear mapping and remains largely constant under circular mapping. Networks with 600 neurons. (**B**) Diffusion increases with network size under linear mapping and decreases under circular mapping. Networks with 3 bumps. (**C** and **D**) Same as **A** and **B**, but for networks with Poisson spiking noise instead of input noise. Dotted gray lines indicate Eq 20. Points indicate data from 48 replicate simulations and bars indicate bootstrapped standard deviations.

As fully explained in the Theoretical model section (Eq 99), we can approximate this spiking process by the rate-based dynamics in Eqs 1 and 2 with the noise term

$$\zeta_\alpha(x, t) = \sum_\beta \int \mathrm{d}y \, W_\beta(x, y) \sqrt{\frac{\phi[g_\beta(y, t)]}{\Delta t}} \eta_\beta(y, t).$$
(18)

The $\eta$'s are independent random variables with zero mean and unit variance:

$$\langle \eta_\alpha(x, t) \rangle = 0, \qquad \langle \eta_\alpha(x, t)\eta_\beta(y, t') \rangle = \Delta t \, \delta(t - t')\delta_{\alpha\beta}\delta(x - y).$$
(19)

As for Eq 9, the simulation timestep $\Delta t$ sets the rate at which $\eta$ is resampled. This spiking noise produces bump diffusion with coefficient (see also Ref [43])

$$D_{\mathrm{spike}} = \frac{\int \mathrm{d}x \, s(x) \left(\frac{\mathrm{d}s}{\mathrm{d}x}\right)^2}{4\tau^2 \left[\int \mathrm{d}x \left(\frac{\mathrm{d}s}{\mathrm{d}x}\right)^2\right]^2}.$$
(20)

As before, $s$ is the baseline firing rate configuration without noise and drive. Through the relationships in Eqs 11 and 12, $D_{\mathrm{spike}}$ scales with $M$ and $N$ in the same way as $D_{\mathrm{input}}$ does:

$$D_{\mathrm{spike}} \propto \frac{N}{M^2} \quad \text{linear}, \qquad D_{\mathrm{spike}} \propto \frac{1}{N} \quad \text{circular}.$$
(21)

These findings are presented in Fig 5C and 5D along with simulation results that confirm our theory. Spiking noise behaves similarly to input noise. Increasing bump number improves robustness to noise under linear mapping but has almost no effect under circular mapping. Bump diffusion in larger networks is exacerbated under linear mapping but suppressed under

circular mapping. For both input noise and spiking noise, the conversion factor in Eq 12 produces the differences between the two mappings. Coupling strength rescaling in Eq 13 does not play a role because $\gamma$ does not appear in Eqs 10 and 20. In S1 Text, we consider splitting a large network with many bumps into smaller networks, each with fewer bumps; the intact network and the combined readout of the split networks exhibit similar diffusion properties.

To evaluate noise robustness in a complementary way, we perform mutual information analysis of networks with input noise. Mutual information describes how knowledge of one random variable can reduce the uncertainty in another, and it serves as a general metric for encoding quality. Before proceeding, we mention a related quantity called Fisher information, which is directly related to mutual information [47, 48] and inversely related to bump diffusion in CANs [43]. Thus, we expect that networks with less diffusion in Fig 5 should generally contain more mutual information about their encoded coordinate. Fisher information also permits a more intuitive explanation for our diffusion scalings. It is proportional to network size $N$ [43], because monitoring a larger number of noisy neurons tells us more about the encoded coordinate. Otherwise, it only depends on the tuning curves of neurons within the network; in particular, steeper tuning curves convey quadratically more information [43]. For our networks, across $M$ and $N$, linear tuning curves are simple rescalings of each other with derivative inversely proportional to $\lambda = N/M$, while circular tuning curves remain unimodal and identical (Figs 2F and 4). Thus, Fisher information should be proportional to $N \cdot (M/N)^2$ and $N$ for linear and circular coordinates, respectively. Applying the inverse proportionality between Fisher information and diffusion coefficients discovered by Ref [43], we arrive at Eqs 15 and 21.

Using simulations, we investigate the mutual information $I$ between the physical coordinate encoded by the noisy network, represented by the random variable $U$ with discretized sample space $\mathcal{U}$, and the activity of a single neuron, represented by the random variable $S$ with discretized sample space $\mathcal{S}$ (see Simulation methods):

$$I[S; U] = \sum_{s \in \mathcal{S}, u \in \mathcal{U}} p(s|u)p(u)\log\frac{p(s|u)}{p(s)}. \tag{22}$$

We then average $I$ across neurons. Larger mean mutual information implies more precise coordinate encoding and greater robustness to noise. Note that the joint activities of all the neurons confer much more coordinate information than single-neuron activities do, but since estimating high-dimensional probability distributions over the former is computationally very costly, we use the latter as a metric for network performance.

The physical coordinate $U$ is either position or orientation and obeys the mappings described in Fig 4 across bump numbers $M$ and network sizes $N$. To obtain the probability distributions in Eq 22 required to compute $I$, we initialize multiple replicate simulations at evenly spaced coordinate values $u$ (Fig 6A). We do not apply a driving input, so the networks should still encode their initialized coordinates at the end of the simulation. However, they contain input noise that degrades their encoding. Collecting the final firing rates produces $p(s|u)$ for each neuron. For both position and orientation, we consider narrow and wide coordinate ranges to assess network performance in both regimes.

We first consider narrow coordinate ranges. For linear coordinates, information increases as a function of $M$ but decreases as a function of $N$; for circular coordinates, it does not strongly depend on $M$ and increases as a function of $N$ (Fig 6B and 6C). These results exactly corroborate those in Fig 5A and 5B obtained for bump diffusion, since we expect information and diffusion to be inversely related.

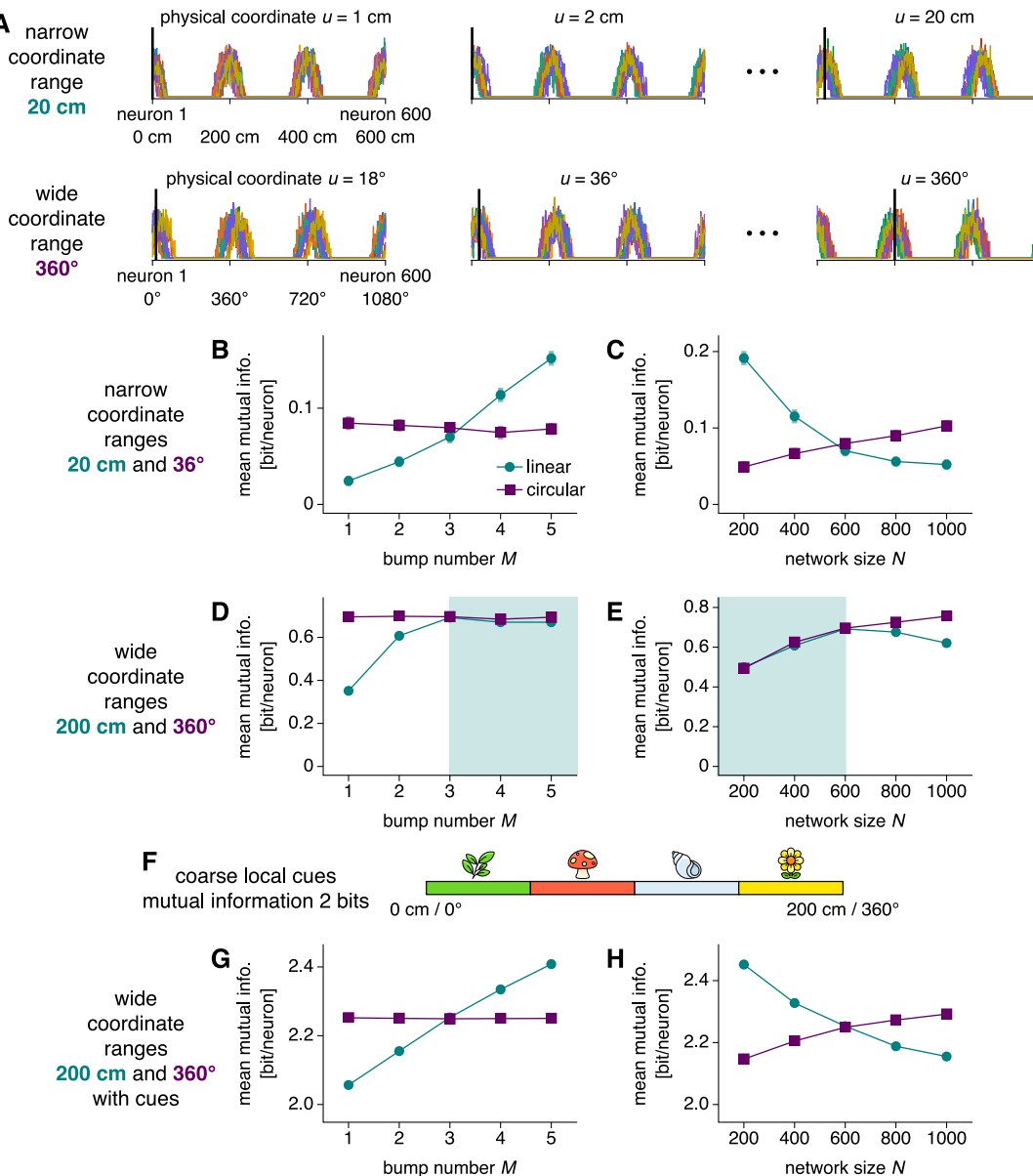

**Fig 6. Mutual information between neural activity and physical coordinates with input noise of magnitude *σ* = 0.5.** (**A**) To compute mutual information, we initialize replicate simulations without input drive at different coordinate values (thick black lines) and record the final neural activities (thin colored lines). The physical coordinate can be linear or circular and its range can be narrow or wide; here, we illustrate two possibilities for networks with 600 neurons and 3 bumps. (**B** and **C**) Mutual information between physical coordinate and single-neuron activity under narrow coordinate ranges. (**B**) Information increases with bump number for linear coordinates and remains largely constant for circular coordinates. Networks with 600 neurons. (**C**) Information decreases with network size for linear coordinates and increases for circular coordinates. Networks with 3 bumps. (**D** and **E**) Mutual information between physical coordinate and single-neuron activity under wide coordinate ranges. The trends in **B** and **C** are preserved for circular coordinates. They are also preserved for linear coordinates, except for the shaded regions in which the coordinate range exceeds the bump distance. (**F**) Coarse local cues are active over different quadrants of the wide coordinate ranges. (**G** and **H**) Mutual information between physical coordinate and the joint activities of a single neuron with the four cues in **F** under wide coordinate ranges. The trends in **B** and **C** are preserved for both linear and circular coordinates. Points indicate data from 96 replicate simulations at each coordinate value averaged over neurons and bars indicate bootstrapped standard errors of the mean. Cue icons adapted from Streamline Freemoji (CC BY license, https://www. streamlinehq.com/emojis/freebies-freemojis).

We next consider wide coordinate ranges. Our ring networks can only uniquely represent coordinate ranges up to their bump distances (converted to physical distances by Fig 4). Beyond these values, two physical coordinates separated by the converted bump distance cannot be distinguished by the network. Our mutual information analysis captures this phenomenon; for linear coordinates, the increase in information with larger $M$ or smaller $N$ as observed in Fig 6B and 6C disappears once the converted bump distance drops below the physical range of 200 cm (green shaded regions of Fig 6D and 6E). In this regime, the benefits of more bumps and smaller networks toward decreasing diffusion are counteracted by bump ambiguity. In contrast, the circular mapping in Fig 4 lacks bump ambiguity since the bump distance is always converted to the maximum physical range of 360˚, so the same qualitative trends in mutual information are observed for any coordinate range (Fig 6D and 6E).

For linear coordinates with wide ranges, the advantages of increasing bump number can be restored by coarse local cues. We illustrate this process by introducing four cues, each of which is active over a different quadrant and is otherwise inactive (Fig 6F). They can be conceptualized as two-state sensory neurons or neural populations that fire in the presence of a local stimulus. By themselves, the cues do not encode precise coordinate values. Mutual information computed with the joint activity of each neuron with these cues recovers the behavior observed for narrow ranges across all $M$ and $N$ (Fig 6G and 6H). Ring attractor neurons provide information beyond the 2 bits conveyed by the cues alone, and for position, this additional information increases with more bumps and fewer neurons without saturating.

In summary, our conclusions about robustness to input noise obtained by diffusion analysis are also supported by mutual information analysis. Moreover, the latter explicitly reveals how networks encoding wide, linear coordinate ranges can leverage coarse local cues to address ambiguities and preserve the advantages of multiple bumps. In S1 Text, we calculate mutual information for a few additional regimes (Fig C of S1 Text; see also Refs [49] and [50], which investigated the Fisher information conveyed by multi-bump tuning curves).

## More bumps improve robustness to connectivity noise under linear mapping

Another source of noise in biological CANs is inhomogeneity in the connectivity $W$. Perfect continuous attractor dynamics requires $W$ to be invariant to translations along the network [9, 10, 16, 18, 28], a concept related to Goldstone's theorem in physics [51, 52]. We consider the effect of replacing $W \to W + V$, where $V$ is a noisy connectivity matrix whose entries are independently drawn from a zero-mean Gaussian distribution. $V$ disrupts the symmetries of $W$. This noise is quenched and does not change over the course of the simulation, in contrast to input and spiking noise, which are independently sampled in time. It contributes a noise term

$$\zeta_\alpha(x,t) = \sum_\beta \int \mathrm{d}y \, V_{\alpha\beta}(x,y) s_\beta(y,t). \tag{23}$$

This formula implies that $V$ produces correlated $\zeta$'s across neurons, which also differs from input and spiking noise. Because of these distinctions, the dominant effect of connectivity noise is not diffusion, but drift. $V$ induces bumps to move with velocity $v_{\mathrm{conn}}(\theta)$, even without drive $b$:

$$v_{\mathrm{conn}}(\theta) = -\frac{\sum_{\alpha\beta} \iint \mathrm{d}x \, \mathrm{d}y \, V_{\alpha\beta}(x,y) \frac{\mathrm{d}s(x-\theta)}{\mathrm{d}x} s(y-\theta)}{2\tau \int \mathrm{d}x \left(\frac{\mathrm{d}s}{\mathrm{d}x}\right)^2}. \tag{24}$$

The movement is coherent but irregular, as it depends on the bump position $\theta$ (Fig 7A). Refs [53] and [54] refer to $v_{\text{conn}}(\theta)$ as the drift velocity.

Connectivity noise traps bumps at low drive $b$. We first consider $b = 0$, so bump motion is governed solely by drift according to $d\theta/dt = v_{\text{conn}}(\theta)$. The bump position $\theta$ has stable fixed points wherever $v_{\text{conn}}(\theta)$ crosses 0 with negative slope [53, 54]. Simulations confirm that bumps drift towards these points (Fig 7B). The introduction of $b$ adds a constant $v_{\text{drive}}$ that moves the curve in Fig 7A up for positive $b$ or down for negative $b$:

$$v_{\text{total}}(\theta) = v_{\text{drive}} + v_{\text{conn}}(\theta). \tag{25}$$

If $v_{\text{total}}(\theta)$ still crosses 0, bumps would still be trapped. The absence of bump motion in response to coordinate changes encoded by $b$ would be a catastrophic failure of path integration. To permit bump motion through the entire network, the drive must be strong enough to eliminate all zero-crossings. Fig 7C shows bump motion at this drive for both directions of motion. The positive $b$ is just large enough for the bump to pass through the region with the most negative $v_{\text{conn}}(\theta)$ in Fig 7A; likewise for negative $b$ and positive $v_{\text{conn}}(\theta)$. We call the larger absolute value of these two drives the escape drive $b_0$. Simulations show that $b_0$ decreases with bump number $M$ and increases with network size $N$ under linear mapping (Fig 7D and 7E). A smaller $b_0$ implies weaker trapping, so smaller networks with more bumps are more resilient against this phenomenon. Under circular mapping, however, $b_0$ demonstrates no significant dependence on $M$ or $N$. We can predict $b_0$ by inverting the relationship in Eq 8

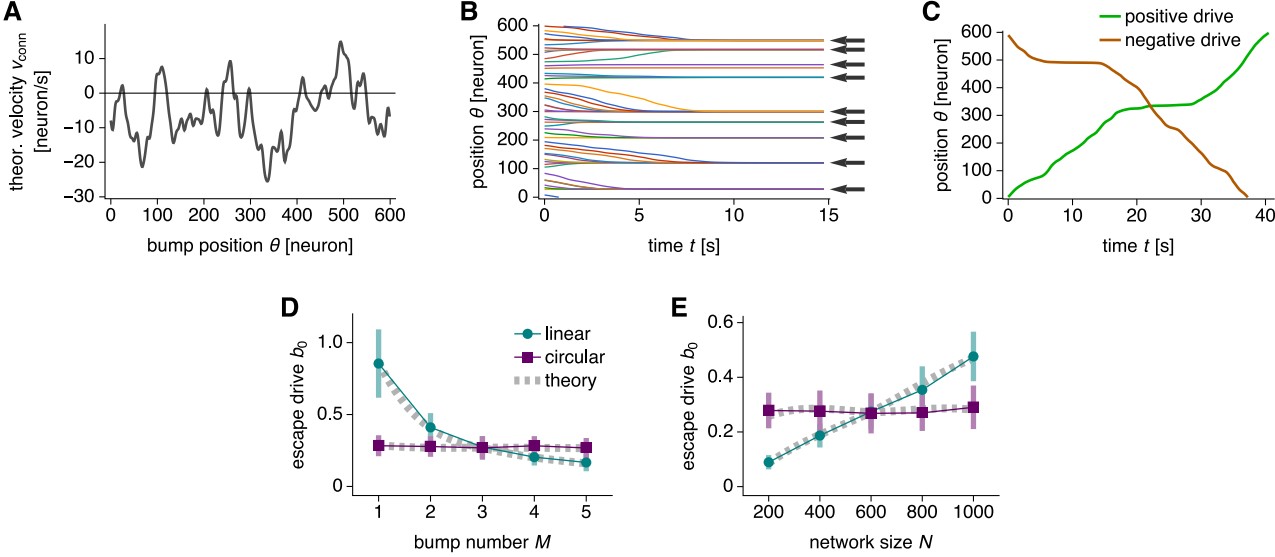

**Fig 7. Bump trapping due to connectivity noise at low drive. (A–C)** Networks with 600 neurons, 1 bump, and the same realization of connectivity noise of magnitude 0.002. **(A)** Theoretical values for drift velocity as a function of bump position using Eq 24. **(B)** Bumps drift towards trapped positions over time. The drive is $b = 0$. Arrows indicate predictions from $v_{\text{conn}}(\theta)$ crossing 0 with negative slope in **A**. Lines indicate simulations with different starting positions. **(C)** Bump trajectories with smallest positive and negative drive required to travel through the entire network. Respectively, $b = 0.75$ and $b = -0.52$. The larger of the two in magnitude is the escape drive $b_0 = 0.75$. Note that positions with low bump speed exhibit large velocities in the opposite direction in **A**. **(D and E)** Networks with multiple realizations of connectivity noise of magnitude 0.002. **(D)** Escape drive decreases with bump number under linear mapping and remains largely constant under circular mapping. Networks with 600 neurons. **(E)** Escape drive increases with network size under linear mapping and remains largely constant under circular mapping. Networks with 3 bumps. Points indicate simulation means over 48 realizations and bars indicate standard deviations. Dotted gray lines indicate Eq 26 averaged over 96 realizations.

between $b$ and $v$:

$$b_0 = -\frac{\max_\theta |v_{\text{conn}}(\theta)| \cdot \tau \int dx \left(\frac{ds}{dx}\right)^2}{\gamma \xi \int dx \frac{d^2 s}{dx^2}}.$$

(26)

This theoretical result agrees well with values obtained by simulation (Fig 7D and 7E). In the Theoretical model section, we present a heuristic argument (Eq 124) that leads to the observed scaling of escape drive on $M$ and $N$:

$$b_0 \propto \frac{N}{M} \quad \text{linear,} \qquad b_0 \propto 1 \quad \text{circular.}$$

(27)

At high drive $|b| > b_0$, attractor bumps are no longer trapped by the drift velocity $v_{\text{conn}}(\theta)$. Instead, the drift term produces irregularities in the total velocity $v_{\text{total}}(\theta)$ (Fig 8A). They can be decomposed into two components: irregularities between directions of motion and irregularities within each direction. Both imply errors in path integration because $v$ and $b$ are not strictly proportional. To quantify these components, we call $|v_+(\theta)|$ and $|v_-(\theta)|$ the observed bump speeds under positive and negative $b$. We define speed difference as the unsigned difference between mean speeds in either direction, normalized by the overall mean speed:

$$\text{speed difference} = \frac{2 \left|\text{mean}_\theta |v_+(\theta)| - \text{mean}_\theta |v_-(\theta)|\right|}{\text{mean}_\theta |v_+(\theta)| + \text{mean}_\theta |v_-(\theta)|}.$$

(28)

We then define speed variability as the standard deviation of speeds within each direction, averaged over both directions and normalized by the overall mean speed:

$$\text{speed variability} = \frac{\text{std}_\theta |v_+(\theta)| + \text{std}_\theta |v_-(\theta)|}{\text{mean}_\theta |v_+(\theta)| + \text{mean}_\theta |v_-(\theta)|}.$$

(29)

Speed difference and speed variability follow the same trends under changes in bump number $M$ and network size $N$ (Fig 8B–8E). Under linear mapping, they decrease with $M$ and increase with $N$. Under circular mapping, they do not significantly depend on $M$ and $N$. These are also the same trends exhibited by the escape drive $b_0$ (Fig 7D and 7E). In terms of theoretical quantities, the formulas for speed difference and variability become

$$\text{speed difference} = \frac{2 \left|\text{mean}_\theta v_{\text{conn}}(\theta)\right|}{|v_{\text{drive}}|}$$

(30)

and

$$\text{speed variability} = \frac{\text{std}_\theta v_{\text{conn}}(\theta)}{|v_{\text{drive}}|}.$$

(31)

These expressions match the observed values well (Fig 8B–8E). In the Theoretical methods section, we calculate the observed dependence of speed difference (Eq 113) and variability (Eq

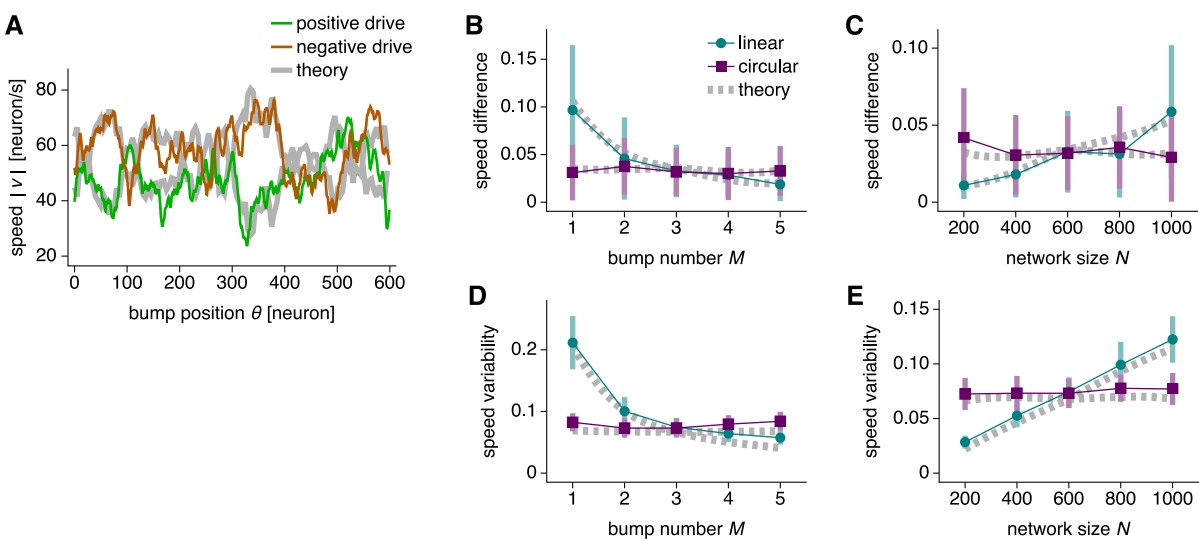

**Fig 8. Bump speed irregularity due to connectivity noise at high drive.** (**A**) Bump speed as a function of bump position with connectivity noise of magnitude 0.002 and drive $b = 1.5$. Network with 600 neurons, 1 bump, and the same realization of connectivity noise as in Fig 7A–7C. Thick gray lines indicate Eq 25. (**B–E**) Networks with multiple realizations of connectivity noise of magnitude 0.002 and drive $b = 1.5$. (**B**) Speed difference between directions decreases with bump number under linear mapping and remains largely constant under circular mapping. Networks with 600 neurons. (**C**) Speed difference increases with network size under linear mapping and remains largely constant under circular mapping. Networks with 3 bumps. (**D and E**) Same as B and C, but for speed variability within each direction. Points indicate simulation means over 48 realizations and bars indicate standard deviations. Dotted gray lines indicate Eqs 30 and 31 averaged over 96 realizations.

120) on $M$ and $N$:

$$
\begin{aligned}
\text{speed difference and variability} \quad &\propto \frac{N}{M} \quad &&\text{linear,} \\
\text{speed difference and variability} \quad &\propto 1 \quad &&\text{circular.}
\end{aligned}
\tag{32}
$$

For all results related to connectivity noise, the coupling strength rescaling in Eq 13 produces the differences between the two mappings via the $\gamma$ in Eq 8. The conversion factor in Eq 12 does not play a role because escape drive, speed difference, and speed variability do not have dimensions of network distance.

To summarize, CANs with imperfect connectivity benefit from more attractor bumps when encoding linear coordinates. This advantage is present at all driving inputs and may be more crucial for larger networks. On the other hand, connectivity noise has largely the same consequences for networks of all bump numbers and sizes when encoding circular coordinates.

## Discussion

We demonstrated how CANs capable of path integration respond to three types of noise. Additive synaptic input noise and Poisson spiking noise cause bumps to diffuse away from the coherent motion responsible for path integration (Figs 3 and 5). This diffusion is accompanied by a decrease in mutual information between neural activity and encoded coordinate (Fig 6). Connectivity noise produces a drift velocity field that also impairs path integration by trapping bumps at low drive and perturbing bump motion at high drive (Figs 7 and 8).

For all three types of noise, CANs with more attractor bumps exhibit less deviation in bump motion in network units. This is observed across network parameters (Figs A and B in

S1 Text). Thus, they can more robustly encode linear variables whose mapping inherits network units and does not rescale with bump number (Fig 4A). If grid cell networks were to encode spatial position in this manner, then multiple attractor bumps would be preferred over a single bump. Ref [20] reports experimental evidence supporting multi-bump grid networks obtained by calcium imaging of mouse medial entorhinal cortex. Our work implies that the evolution of such networks may have been partially encouraged by biological noise. Additional bumps introduce greater ambiguity among positions encoded by each bump, but this can be resolved by a rough estimate of position from additional cues, such as local landmarks [44, 55–57], another grid module with different periodicity [15, 40, 41, 44, 46, 58], or a Bayesian prior based on recent memory [45]. In this way, grid modules with finer resolution and more attractor bumps could maintain a precise egocentric encoding of position, while coarser modules and occasional allocentric cues would identify the true position out of the few possibilities represented. We explicitly explored one realization of this concept and observed how cues enable networks to improve their information content by increasing bump number, despite a concomitant increase in bump ambiguity (Fig 6F–6H).

In contrast, CANs encoding circular variables may rescale under different bump numbers to match periodicities (Fig 4B), which eliminates any influence of bump number on encoding accuracy for all three types of noise. If head direction networks were to encode orientation in this manner, then they would face less selective pressure to evolve beyond the single-bump configuration observed in *Drosophila* [8]. Moreover, without the assumption of bump shape invariance accomplished by Eq 7, robustness to all three types of noise decreases with bump number, which actively favors single-bump orientation networks (Fig B in S1 Text). Further experimental characterization of bump number in biological CANs, perhaps through techniques proposed by Ref [59], would test the degree to which the brain can leverage the theoretical advantages identified in this work.

Under linear mapping, larger CANs exhibit more errors in path integration from all three types of noise. The immediate biological implication is that larger brains face a dramatic degradation of CAN performance, accentuating the importance of suppressing error with multibump configurations. However, the simple rule that one neuron always represents a fixed physical interval does not need to be followed, and larger animals may tolerate greater absolute errors in path integration because they interact with their environments over larger scales. Nevertheless, our results highlight the importance of considering network size when studying the performance of noisy CANs. Under circular mapping, bump diffusion decreases with network size for input and spiking noise, and the magnitude of errors due to connectivity noise is independent of network size. This implies that head direction networks can benefit from incorporating more neurons; the observed interactions among such networks across different mammalian brain regions may act in this manner to suppress noise [60].

The computational advantages of periodic over nonperiodic encodings has been extensively studied in the context of grid cells [45, 46, 49, 61–64]. Our results extend these findings by demonstrating that some kinds of periodic encodings can perform better than others. Our results also contribute to a rich literature on noisy CANs. Previous studies have investigated additive input noise [35, 43, 54, 65, 66], multiplicative input noise [67], spiking noise [18, 43, 54, 62, 63, 68], and quenched noise due to connectivity or input inhomogeneities [10, 53, 54, 66, 69]. Among these works, the relationship between bump number and noise has only been considered in the context of multiple-item working memory, in which a single network can be dynamically loaded with different numbers of bumps [62, 63, 67, 68]. Interestingly, they find that robustness to noise decreases with bump number, which is opposite to our results (cf. Ref [63], which reports no dependence between bump number and encoding accuracy under certain conditions). It appears that CANs designed for path integration with fixed bump number

and CANs designed for multiple-item working memory with variable bump number differ crucially in their responses to noise. Further lines of investigation that compare these two classes would greatly contribute to our general understanding of CANs.

Beyond our concrete results on CAN performance, our work offers a comprehensive theoretical framework for studying path-integrating CANs. We derive a formula for the multi-bump attractor state and a Lyapunov functional that governs its formation. For a ReLU activation function, we calculate all key dynamical quantities such as velocities and diffusion coefficients in terms of firing rates. Our formulas yield scaling relationships that facilitate an intuitive understanding for their dependence on bump number and network size. Much of our theoretical development does not assume a specific connectivity matrix or nonlinear activation function, which allows our results to have wide significance. For example, we expect them to hold for path-integrating networks that contain excitatory synapses. Other theories have been developed for bump shape [37, 38, 53, 66, 67, 70], path integration velocity [37, 42], diffusion coefficients [35, 43, 54, 66, 67, 71], and drift velocity [10, 53, 54]. Our work unifies these studies through a simple framework that features path integration, multiple bumps, and a noise term that can represent a wide range of sources. It can be easily extended to include other components of theoretical or biological significance, such as slowly-varying inputs [27, 66, 72], synaptic plasticity [34, 73], neural oscillations [74–76], and higher-dimensional attractor manifolds [2, 28].

## Theoretical model

### Architecture

We investigate CAN dynamics through a 1D ring attractor network. This class of network has been analyzed in previous theoretical works, and at various points, our calculations will parallel those in Refs [37, 38, 43, 53, 54, 66], and [42].

There are two neurons at each position $i = 0, \ldots, N − 1$ with population indexed by $\alpha \in \{L, R\}$ (Fig 1A). For convenient calculation, we unwrap the ring and connect copies end-to-end, forming a linear network with continuous positions $x \in (−\infty, \infty)$. Unless otherwise specified, integrals are performed over the entire range. To map back onto the finite-sized ring network, we enforce our results to have a periodicity $\lambda$ that divides $N$. For example, $\lambda = N$ corresponds to a single-bump configuration. Integrals would then be performed over $[0, N)$, with positions outside this range corresponding to their equivalents within this range.

The network obeys the following dynamics for synaptic inputs $g$:

$$\tau \frac{\mathrm{d}g_\alpha(x,t)}{\mathrm{d}t} + g_\alpha(x,t) = \sum_\beta \int \mathrm{d}y\, W_\beta(x,y)s_\beta(y,t) + A \pm_\alpha \gamma b(t) + \zeta_\alpha(x,t), \qquad (33)$$

where $\pm_L$ means $−$ and $\pm_R$ means $+$, and the opposite for $\mp_\alpha$. $\tau$ is the neural time constant, $W$ is the synaptic connectivity, and $A$ is the resting input. The nonlinear activation function $\phi$ converts synaptic inputs to firing rates:

$$s_\alpha(x,t) = \phi[g_\alpha(x,t)]. \qquad (34)$$

Most of our results will apply to general $\phi$, but we also consider a ReLU activation function specifically:

$$\phi[g] = \begin{cases} g & g > 0 \\ 0 & g \leq 0. \end{cases} \qquad (35)$$

In this section, we will explicitly mention when we specifically consider the ReLU case, and we will always simplify the function away. Thus, if an expression contains the symbol $\phi$, then it holds for general $\phi$. In the Results section, formulas for $D_{\text{input}}$, $D_{\text{spike}}$, $v_{\text{drive}}$, and $v_{\text{conn}}(\theta)$ as well as all simulation results invoke Eq 35. We will also use this form in the Bump shape $g$ subsection of this section. On the other hand, scalings for $D_{\text{input}}$, $D_{\text{spike}}$, $v_{\text{drive}}$, and $v_{\text{conn}}(\theta)$ in the Results section will hold for general $\phi$, as long as the connectivity obeys Eq 7 such that $g(x/\lambda)$ remains invariant over $M$ and $N$ (Fig 2F). $b$ is the driving input, $\gamma$ is its coupling strength, and $\zeta$ is the noise, which can take different forms. $\gamma b$ and $\zeta$ are small compared to the rest of the right-hand side of Eq 33. For notational convenience, we will often suppress dependence on $t$.

$W_\beta(x, y)$ obeys a standard continuous attractor architecture based on a symmetric and translation-invariant $W$:

$$W_\beta(x, y) = W(x - y \mp_\beta \xi) \quad \text{where} \quad W(-x) = W(x). \tag{36}$$

Each population $\beta$ deviates from $W$ by a small shift $\xi \ll N$ in synaptic outputs. Thus, the following approximation holds:

$$\sum_\beta W_\beta(x, y) \approx 2W(x - y). \tag{37}$$

We will consider the specific form of $W$ (Fig 1B):

$$W(x) = \begin{cases} w \cdot \dfrac{\cos \pi x/l - 1}{2} & |x| < 2l \\ 0 & |x| \geq 2l \end{cases} = \begin{cases} w \cdot \dfrac{\cos kx - 1}{2} & |x| < 2\pi/k \\ 0 & |x| \geq 2\pi/k, \end{cases} \tag{38}$$

where $k = \pi/l$. We will explicitly mention when we specifically consider this form; in fact, we only do so for Eqs 46, 47, 59 and 60, as well as for our simulation results in the Results section. Otherwise, each expression holds for general $W$.

## Baseline configuration without drive and noise

**Linearized dynamics and bump distance $\lambda$.** First, we consider the case of no drive $b = 0$ and no noise $\zeta = 0$. The dynamical equation Eq 33 becomes

$$\tau \frac{\mathrm{d}g_\alpha(x)}{\mathrm{d}t} + g_\alpha(x) = \sum_\beta \int \mathrm{d}y \, W_\beta(x, y) \phi[g_\beta(y)] + A. \tag{39}$$

Since the right-hand side no longer depends on $\alpha$, $g$ must be the same for both populations, and we can use Eq 37 to obtain

$$\tau \frac{\mathrm{d}g(x)}{\mathrm{d}t} + g(x) = 2 \int \mathrm{d}y \, W(x - y) \phi[g(y)] + A. \tag{40}$$

We analyze these dynamics using the Fourier transform $\mathcal{F}$. Our chosen convention, applied to the function $h$, is

$$\tilde{h}(q) = \mathcal{F}[h](q) = \int \mathrm{d}x \, \mathrm{e}^{-iqx} h(x)$$

$$h(x) = \mathcal{F}^{-1}[\tilde{h}](x) = \int \frac{\mathrm{d}q}{2\pi} \, \mathrm{e}^{iqx} \tilde{h}(q). \tag{41}$$

Fourier modes $\tilde{h}(q)$ represent sinusoids with wavenumber $q$ and corresponding wavelength $2\pi/q$. Applying this transform to Eq 40, we obtain

$$\tau \frac{d\tilde{g}(q)}{dt} + \tilde{g}(q) = 2\tilde{W}(q)\mathcal{F}[\phi[g]](q) + 2\pi A\delta(q). \tag{42}$$

In this subsection, we consider the case of small deviations, such that $g(x) \approx g_0$ and $\phi[g(x)] \approx \phi[g_0] + \phi'[g_0](g(x) - g_0)$. Then, Eq 42 becomes

$$\tau \frac{d\tilde{g}(q)}{dt} + \tilde{g}(q) = 2\phi'[g_0]\tilde{W}(q)\tilde{g}(q) + 2\pi A_0\delta(q), \tag{43}$$

where $A_0 = A + 2\tilde{W}(0)(\phi[g_0] - \phi'[g_0]g_0)$. The solution to this linearized equation for $q \neq 0$ is

$$\tilde{g}(q, t) = \tilde{g}(q, 0)e^{r(q)t}. \tag{44}$$

Each mode grows exponentially with rate

$$r(q) = \frac{2\phi'[g_0]\tilde{W}(q) - 1}{\tau}, \tag{45}$$

so the fastest-growing component of $g$ is the one that maximizes $\tilde{W}(q)$, as stated in Eq 5 of the Results section. The wavelength $2\pi/q$ of that component predicts the bump distance $\lambda$.

For the specific $W$ in Eq 38, its Fourier transform is

$$\tilde{W}(q) = -w \frac{k^2 \sin \frac{2\pi q}{k}}{k^2 q - q^3}, \tag{46}$$

so

$$\lambda = \frac{2\pi}{\underset{q}{\operatorname{argmin}} \dfrac{k^2 \sin \frac{2\pi q}{k}}{k^2 q - q^3}} = \frac{2l}{\underset{\psi}{\operatorname{argmin}} \dfrac{\sin 2\pi\psi}{\psi - \psi^3}} \approx 2.28l. \tag{47}$$

$\lambda$ is proportional to $l$, as also noted by Refs [16, 18, 41], and [40].

**Bump shape $g$.**  We call the steady-state synaptic inputs $g$ without drive and noise the baseline configuration. To calculate its shape, we must account for the nonlinearity of the activation function $\phi$ and return to Eq 42. We invoke our particular form of $\phi$ in Eq 35 to calculate $\mathcal{F}[\phi[g]](q)$. $g$ must be periodic, and its periodicity is the bump distance $\lambda$ with wavenumber $\kappa = 2\pi/\lambda$. Without loss of generality, we take $g$ to have a bump centered at 0. Since $W$ is symmetric, $g$ is an even function. We define $z$ as the position where $g$ crosses 0:

$$g(z) = 0. \tag{48}$$

If $g$ is approximately sinusoidal, then $g(x) > 0$ wherever $n\lambda - z < x < n\lambda + z$ for any integer $n$. The ReLU $\phi$ in Eq 35 implies

$$\phi[g(x)] = g(x)\Phi(x) \quad \text{where} \quad \Phi(x) = \sum_{n=-\infty}^{\infty} \Theta[x - n\lambda + z]\Theta[-(x - n\lambda - z)]. \tag{49}$$

$\Theta$ is the Heaviside step function. The Fourier transform for $\Phi$ is

$$\tilde{\Phi}(q) = 2\frac{\sin qz}{q} \sum_{n=-\infty}^{\infty} e^{-2\pi inq/\kappa} = 2\frac{\sin qz}{q} \sum_{n=-\infty}^{\infty} \delta\left(n - \frac{q}{\kappa}\right) = 2\kappa \frac{\sin qz}{q} \sum_{n=-\infty}^{\infty} \delta(q - n\kappa), \tag{50}$$

where the second equality comes from the Fourier series for a Dirac comb. Therefore,

$$\mathcal{F}[\phi[g]](q) = \frac{1}{2\pi} \int dq' \, \tilde{\Phi}(q - q')\tilde{g}(q') = \frac{1}{\pi} \sum_{n=-\infty}^{\infty} \frac{\sin n\kappa z}{n} \tilde{g}(q - n\kappa),$$ (51)

so Eq 42 becomes

$$\tau \frac{d\tilde{g}(q)}{dt} + \tilde{g}(q) = \frac{2}{\pi} \tilde{W}(q) \sum_{n=-\infty}^{\infty} \frac{\sin n\kappa z}{n} \tilde{g}(q - n\kappa) + 2\pi A\delta(q).$$ (52)

This equation describes the full dynamics of $g$ with a ReLU activation function. It contains couplings between all modes $q$ that are multiples of the wavenumber $\kappa$, which corresponds to the bump distance.

To find the baseline $g$, we set $d\tilde{g}/dt = 0$. We also simplify $\tilde{g}(q)$ by only considering the lowest modes that couple to each other: $q = 0, \pm\kappa$. Due to symmetry, $\tilde{W}(-q) = \tilde{W}(q)$ and $\tilde{g}(-q) = \tilde{g}(q)$. Eq 52 gives

$$\begin{aligned} \tilde{g}(0) &= \frac{2}{\pi} \tilde{W}(0)[\kappa z \, \tilde{g}(0) + 2\sin(\kappa z) \, \tilde{g}(\kappa)] + 2\pi A\delta(0) \\ \tilde{g}(\kappa) &= \frac{2}{\pi} \tilde{W}(\kappa) \left[ \sin(\kappa z) \, \tilde{g}(0) + \left( \kappa z + \frac{\sin 2\kappa z}{2} \right) \tilde{g}(\kappa) \right]. \end{aligned}$$ (53)

Now we need to impose Eq 48: $g(z) = 0$. To do so, we note that $\tilde{g}(0)$ and $\tilde{g}(\kappa)$ are both proportional to $\delta(0)$ according to Eq 53. That means $\tilde{g}(q)$ has the form

$$\tilde{g}(q) = G_0\delta(q) + G\delta(q - \kappa) + G\delta(q + \kappa),$$ (54)

where $G_0$ and $G$ are the Fourier modes with delta functions separated. This implies

$$g(x) = \frac{G_0}{2\pi} + \frac{G}{\pi}\cos \kappa x,$$ (55)

and $g(z) = 0$ implies

$$G_0 = -2\cos(\kappa z)\, G.$$ (56)

Substituting Eqs 54 and 56 into Eq 53, we obtain

$$\begin{aligned} \frac{G}{\pi} &= -\frac{\pi A}{2\tilde{W}(0)(\sin \kappa z - \kappa z \cos \kappa z) + \pi \cos \kappa z} \\ \kappa z - \cos \kappa z \sin \kappa z &= \frac{\pi}{2\tilde{W}(\kappa)}. \end{aligned}$$ (57)

We can solve the second equation of Eq 57 for $\kappa z$ and then substitute it into the first equation to obtain $G$. This then gives us $g(x)$, which becomes through Eqs 55 and 56

$$g(x) = \frac{G}{\pi}(\cos \kappa x - \cos \kappa z).$$ (58)

In particular, let's use the $W$ defined by Eq 38 with Fourier transform Eq 46. Then,

$$\tilde{W}(0) = -\frac{2\pi w}{k} \qquad \text{and} \qquad \tilde{W}(\kappa) = -\frac{w}{\kappa} \frac{k^2}{k^2 - \kappa^2} \sin \frac{2\pi\kappa}{k}.$$ (59)

Thus,

$$
\begin{aligned}
g(x) &= \frac{G}{\pi}(\cos \kappa x - \cos \kappa z) \\
\kappa z - \cos \kappa z \sin \kappa z &= -\pi \left/ \left[ \frac{2w}{\kappa} \frac{k^2}{k^2 - \kappa^2} \sin \frac{2\pi\kappa}{k} \right] \right. \\
\frac{G}{\pi} &= A \left/ \left[ \frac{4w}{k}(\sin \kappa z - \kappa z \cos \kappa z) - \cos \kappa z \right] \right. .
\end{aligned}
\tag{60}
$$

This provides expressions for $a$ and $d$ in Eq 4 of the Results section, where $a = G/\pi$ and $d = -(G/\pi) \cos \kappa z$.

**Lyapunov functional and bump distance λ.** The dynamical equation in Eq 40 admits a Lyapunov functional. In analogy to the continuous Hopfield model [77], we can define a Lyapunov functional in terms of $s(x) = \phi[g(x)]$:

$$
L = - \iint \mathrm{d}x \, \mathrm{d}y \, W(x - y)s(x)s(y) + \int \mathrm{d}x \int_0^{s(x)} \mathrm{d}\rho \, \phi^{-1}[\rho] - A \int \mathrm{d}x \, s(x).
\tag{61}
$$

The nonlinearity $\phi$ must be invertible in the range $(0, s)$ for any possible firing rate $s$. For $L$ to be bounded from below for a network of any size $N$, we need

1. $W(x)$ to be negative-definite, and

2. $\int_0^s \mathrm{d}\rho \, \phi^{-1}[\rho] - As$ to be bounded from below for any possible firing rate $s$.

We can check that these hold for our particular functions. Eq 38 immediately shows that the first condition is met. Eq 35 states that $\phi^{-1}[\rho] = \rho$ when $\rho > 0$, so $\int_0^s \mathrm{d}\rho \, \phi^{-1}[\rho] - As = \frac{1}{2}s^2 - As$, which satisfies the second condition.

Now we take the time derivative and use Eq 40:

$$
\begin{aligned}
\frac{\mathrm{d}L}{\mathrm{d}t} &= - \int \mathrm{d}x \left\{ 2 \int \mathrm{d}y \, W(x - y)s(y) - \phi^{-1}[s(x)] + A \right\} \frac{\mathrm{d}s(x)}{\mathrm{d}t} \\
&= -\tau \int \mathrm{d}x \, \frac{\mathrm{d}g(x)}{\mathrm{d}t} \frac{\mathrm{d}s(x)}{\mathrm{d}t} \\
&= -\tau \int \mathrm{d}x \, \phi'[g(x)] \left( \frac{\mathrm{d}g(x)}{\mathrm{d}t} \right)^2 .
\end{aligned}
\tag{62}
$$

As long as $\phi$ is a monotonically nondecreasing function, $\mathrm{d}L/\mathrm{d}t \leq 0$. Thus, $L$ is a Lyapunov functional.

Now we seek to simplify Eq 61. Suppose we are very close to a steady-state solution, so $\mathrm{d}g/\mathrm{d}t \approx 0$. We substitute Eq 40 into Eq 61 to obtain

$$
\begin{aligned}
L &= -\frac{1}{2} \int \mathrm{d}x \, [g(x) - A]s(x) + \int \mathrm{d}x \int_0^{s(x)} \mathrm{d}\rho \, \phi^{-1}[\rho] - A \int \mathrm{d}x \, s(x) \\
&= -\frac{1}{2} \int \mathrm{d}x \, g(x)s(x) + \int \mathrm{d}x \int_0^{s(x)} \mathrm{d}\rho \, \phi^{-1}[\rho] - \frac{A}{2} \int \mathrm{d}x \, s(x).
\end{aligned}
\tag{63}
$$

Now we invoke our ReLU $\phi$ from Eq 35 to obtain

$$L = -\frac{1}{2}\int dx\,[g(x) - s(x)]s(x) - \frac{A}{2}\int dx\,s(x) = -\frac{A}{2}\int dx\,s(x). \tag{64}$$

The last equality was obtained by noticing that for any $x$, either $s(x) = 0$ or $g(x) - s(x) = 0$ with our $\phi$. Therefore, the stable solution that minimizes $L$ is the one that maximizes the total firing rate.

We can apply our sinusoidal $g$ in Eq 58 to perform the integral, recalling that $\kappa = 2\pi/\lambda$:

$$L = -\frac{NAG}{2\pi^2}(\sin\kappa z - \kappa z\cos\kappa z), \tag{65}$$

where $N$ is the network size. So $L$ depends on $G$ and the quantity $\kappa z$, which we will rewrite as $\psi$. We now simplify Eq 65 using Eq 57:

$$L = -\frac{NA^2(\sin\psi - \psi\cos\psi)}{4\tilde{W}(0)(\sin\psi - \psi\cos\psi) - 2\pi\cos\psi} = \frac{NA^2}{-4\tilde{W}(0) + \frac{2\pi}{\tan\psi - \psi}}. \tag{66}$$

Note that since $W$ is negative-definite, $\tilde{W}(0) = \int dx\,W(x) < 0$. Also note that $1/(\tan\psi - \psi)$ is a monotonically decreasing function of $\psi$ in the range $[0, \pi]$. Thus, to minimize $L$, we need to minimize $\psi$. Meanwhile, Eq 57 now reads $\psi - \cos\psi\sin\psi = \pi/2\tilde{W}(\kappa)$. The left-hand side is a monotonically increasing function of $\psi$ in the range $[0, \pi]$, so to minimize $\psi$, we need to maximize $\tilde{W}(\kappa)$. Thus, the Lyapunov-stable wavelength $\lambda = 2\pi/\kappa$ is the one that maximizes $\tilde{W}(\kappa)$. This is the same mode that grows fastest for the linearized dynamics in Eq 45.

## Bump motion under drive and noise

**Dynamics along the attractor manifold.** Now that we have determined the baseline configuration $g$, including the bump shape and bump distance, we investigate its motion under drive $b$ and noise $\zeta$. We introduce $\theta$ to label the position of the configuration. It can be defined as the center of mass or the point of maximum activity of one of the bumps. We expand the full time-dependent configuration with respect to the baseline configuration located at $\theta$:

$$g_\alpha(x, t) = g(x - \theta) + \delta g_\alpha(x, t). \tag{67}$$

$g(x - \theta)$ solves Eq 40 with $dg/dt = 0$; to facilitate calculations below, we will write the baseline equation in this form:

$$g(x - \theta) = \sum_\beta \int dy\,W_\beta(x, y)\phi[g(y - \theta)] + A. \tag{68}$$

Substituting Eq 67 into Eq 33 and invoking Eq 68, we obtain the following linearized dynamics for $\delta g$:

$$\tau\frac{d\delta g_\alpha(x, t)}{dt} + \delta g_\alpha(x, t) = \sum_\beta \int dy\,W_\beta(x, y)\phi'[g(y - \theta)]\delta g_\beta(y, t) \pm_\alpha \gamma b(t) + \zeta_\alpha(x, t). \tag{69}$$

We can rewrite this as

$$\tau\frac{d\delta g_\alpha(x, t)}{dt} = \sum_\beta \int dy\,K_{\alpha\beta}(x, y; \theta)\delta g_\beta(y, t) \pm_\alpha \gamma b(t) + \zeta_\alpha(x, t), \tag{70}$$

where

$$K_{\alpha\beta}(x, y; \theta) = W_{\beta}(x, y)\phi'[g(y - \theta)] - \delta_{\alpha\beta}\delta(x - y). \tag{71}$$

We will often suppress the argument of derivatives of $g$. If we consider a configuration located at $\theta$, $\mathrm{d}g/\mathrm{d}x$ implies $\mathrm{d}g(x - \theta)/\mathrm{d}x$. We make the argument explicit when necessary.

If we differentiate Eq 68 by $\theta$, we obtain

$$\begin{aligned}
\frac{\mathrm{d}g}{\mathrm{d}x} &= \sum_{\beta} \int \mathrm{d}y\, W_{\beta}(x, y)\phi'[g(y - \theta)]\frac{\mathrm{d}g}{\mathrm{d}y} \\
0 &= \sum_{\beta} \int \mathrm{d}y\, K_{\alpha\beta}(x, y; \theta)\frac{\mathrm{d}g}{\mathrm{d}y},
\end{aligned} \tag{72}$$

which indicates that $\mathrm{d}g/\mathrm{d}x$ is a right eigenvector of $K$ with eigenvalue 0. To be explicit about this, we recover the discrete case by converting continuous functions to vectors and matrices:

$$g_i = g(i - \theta), \qquad \Delta g_i = \left.\frac{\mathrm{d}g(x - \theta)}{\mathrm{d}x}\right|_{x=i}, \qquad K_{\alpha\beta ij} = K_{\alpha\beta}(i, j; \theta). \tag{73}$$

If we concatenate matrices and vectors across populations as

$$\mathsf{J} = \begin{pmatrix} \mathsf{K}_{\mathrm{LL}} & \mathsf{K}_{\mathrm{LR}} \\ \mathsf{K}_{\mathrm{RL}} & \mathsf{K}_{\mathrm{RR}} \end{pmatrix}, \qquad \mathbf{e} = \begin{pmatrix} \Delta\mathbf{g} \\ \Delta\mathbf{g} \end{pmatrix}, \tag{74}$$

$\mathbf{e}$ is the right null eigenvector of $\mathsf{J}$: $0 = \sum_j J_{ij} e_j$.

Since $K$ is not symmetric, its left and right eigenvectors may be different. To find the left null eigenvector, we again differentiate Eq 68 with respect to $\theta$, but this time interchanging variables $x$ and $y$:

$$\begin{aligned}
\frac{\mathrm{d}g}{\mathrm{d}y} &= \sum_{\beta} \int \mathrm{d}x\, W_{\beta}(y, x)\phi'[g(x - \theta)]\frac{\mathrm{d}g}{\mathrm{d}x} \\
&\approx 2 \int \mathrm{d}x\, W(x - y)\phi'[g(x - \theta)]\frac{\mathrm{d}g}{\mathrm{d}x}.
\end{aligned} \tag{75}$$

The second equality is obtained from Eqs 36 and 37. Replacing the position $y$ by $y \pm_{\beta} \xi$, where $\xi$ is the connectivity shift, we get

$$\frac{\mathrm{d}g(y - \theta \pm_{\beta} \xi)}{\mathrm{d}y} \approx 2 \int \mathrm{d}x\, W(x - y \mp_{\beta} \xi)\phi'[g(x - \theta)]\frac{\mathrm{d}g(x - \theta)}{\mathrm{d}x}, \tag{76}$$

where we have made the arguments of $g$ explicit. Let's define shifted versions of the baseline $g$ for each population $\alpha$:

$$\bar{g}_{\alpha}(x) = g(x \pm_{\alpha} \xi). \tag{77}$$

Since $\xi$ is small,

$$\sum_{\alpha} \bar{g}_{\alpha}(x) \approx 2g(x). \tag{78}$$

Applying these expressions to Eq 76 and recalling Eq 36,

$$
\begin{aligned}
\frac{\mathrm{d}\bar{g}_\beta}{\mathrm{d}y} &\approx 2 \int \mathrm{d}x\, W_\beta(x, y) \phi'[g(x - \theta)] \frac{\mathrm{d}g}{\mathrm{d}x} \\
&\approx \sum_\alpha \int \mathrm{d}x\, W_\beta(x, y) \phi'[g(x - \theta)] \frac{\mathrm{d}\bar{g}_\alpha}{\mathrm{d}x}.
\end{aligned}
\tag{79}
$$

Finally, we multiply both sides of the equation by $\phi'[g(y - \theta)]$ to obtain

$$
\begin{aligned}
\phi'[g(y - \theta)] \frac{\mathrm{d}\bar{g}_\beta}{\mathrm{d}y} &\approx \sum_\alpha \int \mathrm{d}x\, W_\beta(x, y) \phi'[g(y - \theta)] \phi'[g(x - \theta)] \frac{\mathrm{d}\bar{g}_\alpha}{\mathrm{d}x} \\
0 &= \sum_\alpha \int \mathrm{d}x\, K_{\alpha\beta}(x, y; \theta) \phi'[g(x - \theta)] \frac{\mathrm{d}\bar{g}_\alpha}{\mathrm{d}x}.
\end{aligned}
\tag{82}
$$

Thus $\phi'[g(x - \theta)]\, \mathrm{d}\bar{g}_\alpha/\mathrm{d}x$ is the left null eigenvector for $K_{\alpha\beta}$. Again, to be explicit, the discrete equivalent is

$$
\mathsf{J} = \begin{pmatrix} \mathsf{K}_{\mathrm{LL}} & \mathsf{K}_{\mathrm{LR}} \\ \mathsf{K}_{\mathrm{RL}} & \mathsf{K}_{\mathrm{RR}} \end{pmatrix}, \qquad
\mathbf{f} = \begin{pmatrix} \phi'[\mathbf{g}] \odot \Delta\bar{\mathbf{g}}_{\mathrm{L}} \\ \phi'[\mathbf{g}] \odot \Delta\bar{\mathbf{g}}_{\mathrm{R}} \end{pmatrix},
\tag{81}
$$

where $\odot$ represents element-wise (Hadamard) multiplication. Then, $\mathbf{f}$ is the left null eigenvector of $\mathsf{J} : 0 = \sum_i J_{ij} f_i$.

We now revisit Eq 67 and assume that $g$ changes such that the bumps slowly move along the attractor manifold:

$$
\begin{aligned}
g_\alpha(x, t) &\approx g(x - \theta(t)), \\
\frac{\mathrm{d}\delta g_\alpha(x, t)}{\mathrm{d}t} = \frac{\mathrm{d}g_\alpha(x, t)}{\mathrm{d}t} &\approx -\frac{\mathrm{d}g(x - \theta(t))}{\mathrm{d}x} \frac{\mathrm{d}\theta}{\mathrm{d}t}.
\end{aligned}
\tag{82}
$$

Again for simplicity, we will often suppress arguments of derivatives of $g$ and dependence on $t$. We return to Eq 70, project it along the left null eigenvector, and apply Eq 82 to obtain

$$
\begin{aligned}
-\tau \frac{\mathrm{d}\theta}{\mathrm{d}t} \sum_\alpha \int \mathrm{d}x\, \phi'[g(x - \theta)] \frac{\mathrm{d}\bar{g}_\alpha}{\mathrm{d}x} \frac{\mathrm{d}g}{\mathrm{d}x} = {} & \gamma b \sum_\alpha \int \mathrm{d}x\, (\pm_\alpha 1) \cdot \phi'[g(x - \theta)] \frac{\mathrm{d}\bar{g}_\alpha}{\mathrm{d}x} \\
& + \sum_\alpha \int \mathrm{d}x\, \phi'[g(x - \theta)] \frac{\mathrm{d}\bar{g}_\alpha}{\mathrm{d}x} \zeta_\alpha(x).
\end{aligned}
\tag{83}
$$

The velocity of bump motion is given by $\mathrm{d}\theta/\mathrm{d}t$. It is

$$
\begin{aligned}
\frac{\mathrm{d}\theta}{\mathrm{d}t} \approx {} & -\frac{\gamma b \sum_\alpha \int \mathrm{d}x\, (\pm_\alpha 1) \cdot \phi'[g(x - \theta)] \dfrac{\mathrm{d}\bar{g}_\alpha(x - \theta)}{\mathrm{d}x}}{2\tau \int \mathrm{d}x\, \phi'[g(x - \theta)] \left( \dfrac{\mathrm{d}g(x - \theta)}{\mathrm{d}x} \right)^2} \\
& -\frac{\sum_\alpha \int \mathrm{d}x\, \phi'[g(x - \theta)] \dfrac{\mathrm{d}\bar{g}_\alpha(x - \theta)}{\mathrm{d}x} \zeta_\alpha(x)}{2\tau \int \mathrm{d}x\, \phi'[g(x - \theta)] \left( \dfrac{\mathrm{d}g(x - \theta)}{\mathrm{d}x} \right)^2},
\end{aligned}
\tag{84}
$$

where we have made the arguments of $g$ explicit. This equation encapsulates all aspects of bump motion for our theoretical model. It includes dependence on both drive $b$ and noise $\zeta$,

the latter of which is kept in a general form. We will proceed by considering specific cases of this equation.

**Path integration velocity $v_{\mathrm{drive}}$ due to driving input $b$.** The noiseless case of Eq 84 with $\zeta_\alpha(x) = 0$ yields the bump velocity due to drive $b$, which is responsible for path integration:

$$v_{\mathrm{drive}} = -\frac{\gamma b \int \mathrm{d}x \, \phi'[g(x-\theta)]\left(\dfrac{\mathrm{d}\bar{g}_{\mathrm{R}}}{\mathrm{d}x} - \dfrac{\mathrm{d}\bar{g}_{\mathrm{L}}}{\mathrm{d}x}\right)}{2\tau \int \mathrm{d}x \, \phi'[g(x-\theta)]\left(\dfrac{\mathrm{d}g}{\mathrm{d}x}\right)^2}. \tag{85}$$

Note that this expression is independent of the position $\theta$. We can explicitly remove $\theta$ by shifting the dummy variable $x \to x + \theta$:

$$
\begin{aligned}
v_{\mathrm{drive}} &= -\frac{\gamma b \int \mathrm{d}x \, \phi'[g(x)]\left(\dfrac{\mathrm{d}g(x+\xi)}{\mathrm{d}x} - \dfrac{\mathrm{d}g(x-\xi)}{\mathrm{d}x}\right)}{2\tau \int \mathrm{d}x \, \phi'[g(x)]\left(\dfrac{\mathrm{d}g(x)}{\mathrm{d}x}\right)^2} \\[2em]
&\approx -\frac{\gamma b \xi \int \mathrm{d}x \, \phi'[g(x)]\dfrac{\mathrm{d}^2 g}{\mathrm{d}x^2}}{\tau \int \mathrm{d}x \, \phi'[g(x)]\left(\dfrac{\mathrm{d}g}{\mathrm{d}x}\right)^2}.
\end{aligned}
\tag{86}
$$

Now let's consider the specific ReLU activation function $\phi$. Eq 35 implies

$$\phi'[g] = \begin{cases} 1 & g > 0 \\ 0 & g \le 0, \end{cases} \quad \text{so} \quad \phi'[g]^2 = \phi'[g] \quad \text{and} \quad \phi'[g] \cdot \phi[g] = \phi[g]. \tag{87}$$

These identities, along with the definition for $s$ (Eq 34), give

$$\phi'[g(x)]\frac{\mathrm{d}^2 g}{\mathrm{d}x^2} = \frac{\mathrm{d}^2 s}{\mathrm{d}x^2}, \qquad \phi'[g(x)]\left(\frac{\mathrm{d}g}{\mathrm{d}x}\right)^2 = \left(\frac{\mathrm{d}s}{\mathrm{d}x}\right)^2, \qquad \phi[g(x)]\left(\frac{\mathrm{d}g}{\mathrm{d}x}\right)^2 = s(x)\left(\frac{\mathrm{d}s}{\mathrm{d}x}\right)^2. \tag{88}$$

Applying the first two equalities to Eq 86 produces Eq 8 of the Results section.

Now we reintroduce noise $\zeta$ and assume it is independent across neurons and timesteps, with mean $\langle\zeta\rangle$. If we average Eq 84 over $\zeta$, the numerator of the second term becomes

$$\sum_\alpha \int \mathrm{d}x \, \phi'[g(x-\theta)]\frac{\mathrm{d}\bar{g}_\alpha(x-\theta)}{\mathrm{d}x}\langle\zeta\rangle = 0. \tag{89}$$

The integral vanishes because $g$ is even and $\sum_\alpha \mathrm{d}\bar{g}_\alpha/\mathrm{d}x$ is odd. Thus,

$$\left\langle\frac{\mathrm{d}\theta}{\mathrm{d}t}\right\rangle = v_{\mathrm{drive}}, \tag{90}$$

demonstrating that networks with independent noise still path integrate on average.

**Diffusion $D_{\mathrm{input}}$ due to input noise.** Independent noise $\zeta$ produces diffusion, a type of deviation in bump motion away from the average trajectory. It is quantified by the diffusion coefficient $D$:

$$\langle[\theta(t) - \langle\theta(t)\rangle]^2\rangle = 2Dt. \tag{91}$$

In terms of derivatives of $\theta$,

$$\langle[\theta(t) - \langle\theta(t)\rangle]^2\rangle = \int_0^t \int_0^t dt' \, dt'' \left\langle \left(\frac{d\theta}{dt'} - \left\langle\frac{d\theta}{dt'}\right\rangle\right)\left(\frac{d\theta}{dt''} - \left\langle\frac{d\theta}{dt''}\right\rangle\right)\right\rangle. \tag{92}$$

Eqs 84 and 90 imply

$$\frac{d\theta}{dt} - \left\langle\frac{d\theta}{dt}\right\rangle = -\frac{\sum_\alpha \int dx \, \phi'[g(x-\theta)]\frac{d\bar{g}_\alpha}{dx}\zeta_\alpha(x)}{2\tau\int dx \, \phi'[g(x-\theta)]\left(\frac{dg}{dx}\right)^2}. \tag{93}$$

We then shift the dummy variable $x \to x + \theta(t)$ and reintroduce explicit dependence on $t$ to obtain

$$\langle[\theta(t) - \langle\theta(t)\rangle]^2\rangle = \int_0^t \int_0^t dt' \, dt'' \frac{\sum_{\alpha\beta} \iint dx \, dy \, \phi'[g(x)]\phi'[g(y)]\frac{d\bar{g}_\alpha}{dx}\frac{d\bar{g}_\beta}{dy}\langle\zeta_\alpha(x+\theta(t'),t')\zeta_\beta(y+\theta(t''),t'')\rangle}{4\tau^2\left[\int dx \, \phi'[g(x)]\left(\frac{dg}{dx}\right)^2\right]^2}. \tag{94}$$

One class of independent $\zeta$ is Gaussian noise added to the total synaptic input, which represents neural fluctuations at short timescales. We assume it is independent across neurons and timesteps with zero mean and fixed variance $\sigma^2$:

$$\langle\zeta_\alpha(x,t)\rangle = 0, \qquad \langle\zeta_\alpha(x,t)\zeta_\beta(y,t')\rangle = \sigma^2\Delta t \, \delta(t-t')\delta_{\alpha\beta}\delta(x-y). \tag{95}$$

$\Delta t$ is the simulation timestep, which defines the rate at which the random noise variable is resampled. Eq 94 then becomes, with the help of Eq 78,

$$\begin{aligned}\langle[\theta(t) - \langle\theta(t)\rangle]^2\rangle &= \int_0^t dt' \frac{\sigma^2\Delta t\sum_\alpha \int dx \, \phi'[g(x)]^2\left(\frac{d\bar{g}_\alpha}{dx}\right)^2}{4\tau^2\left[\int dx \, \phi'[g(x)]\left(\frac{dg}{dx}\right)^2\right]^2}\\[2em] &\approx \frac{\sigma^2\Delta t\int dx \, \phi'[g(x)]^2\left(\frac{dg}{dx}\right)^2}{2\tau^2\left[\int dx \, \phi'[g(x)]\left(\frac{dg}{dx}\right)^2\right]^2} \cdot t.\end{aligned} \tag{96}$$

Reconciling this with the definition of the diffusion coefficient $D$ in Eq 91 yields

$$D_{\text{input}} = \frac{\sigma^2\Delta t\int dx \, \phi'[g(x)]^2\left(\frac{dg}{dx}\right)^2}{4\tau^2\left[\int dx \, \phi'[g(x)]\left(\frac{dg}{dx}\right)^2\right]^2}. \tag{97}$$

Applying Eq 88 for a ReLU $\phi$ gives Eq 10 of the Results section.

**Diffusion $D_{\text{spike}}$ due to spiking noise.** Instead of input noise, we consider independent noise arising from spiking neurons. In this case, the stochastic firing rate $s$ is no longer the

deterministic expression in Eq 34. Instead,

$$s_\alpha(x, t) = \frac{c_\alpha(x, t)}{\Delta t}, \tag{98}$$

where $c$ is the number of spikes emitted in a simulation timestep of length $\Delta t$. We model each $c_\alpha(x, t)$ as an independent Poisson-like random variable driven by the deterministic firing rate $\phi[g_\alpha(x, t)]$ with Fano factor $F$. It has mean $\phi[g_\alpha(x, t)]\Delta t$ and variance $F\phi[g_\alpha(x, t)]\Delta t$. Therefore,

$$s_\alpha(x, t) = \phi[g_\alpha(x, t)] + \sqrt{\frac{F\phi[g_\alpha(x, t)]}{\Delta t}}\eta_\alpha(x, t), \tag{99}$$

where each $\eta_\alpha(x, t)$ is an independent random variable with zero mean and unit variance:

$$\langle \eta_\alpha(x, t) \rangle = 0, \qquad \langle \eta_\alpha(x, t)\eta_\beta(y, t') \rangle = \Delta t\, \delta(t - t')\delta_{\alpha\beta}\delta(x - y). \tag{100}$$

As in Eq 95, the simulation timestep $\Delta t$ defines the rate at which $\eta$ is resampled. By substituting Eq 99 into Eq 33, we see that spiking neurons can be described by deterministic firing rate dynamics with the stochastic noise term

$$\zeta_\alpha(x, t) = \sum_\beta \int \mathrm{d}y\, W_\beta(x, y)\sqrt{\frac{F\phi[g_\beta(y, t)]}{\Delta t}}\eta_\beta(y, t). \tag{101}$$

Now we calculate the diffusion coefficient produced by this noise. Eq 93 becomes

$$\begin{aligned}
\frac{\mathrm{d}\theta}{\mathrm{d}t} - \left\langle \frac{\mathrm{d}\theta}{\mathrm{d}t} \right\rangle &= -\frac{\displaystyle\sum_{\alpha\beta} \iint \mathrm{d}x\,\mathrm{d}y\, W_\beta(x, y)\phi'[g(x - \theta)]\frac{\mathrm{d}\bar{g}_\alpha}{\mathrm{d}x}\sqrt{\frac{F\phi[g(y - \theta)]}{\Delta t}}\eta_\beta(y)}{2\tau \displaystyle\int \mathrm{d}x\, \phi'[g(x - \theta)]\left(\frac{\mathrm{d}g}{\mathrm{d}x}\right)^2} \\
&= -\frac{\displaystyle\sum_\beta \int \mathrm{d}y\, \frac{\mathrm{d}\bar{g}_\beta}{\mathrm{d}y}\sqrt{\frac{F\phi[g(y - \theta)]}{\Delta t}}\eta_\beta(y)}{2\tau \displaystyle\int \mathrm{d}x\, \phi'[g(x - \theta)]\left(\frac{\mathrm{d}g}{\mathrm{d}x}\right)^2}.
\end{aligned} \tag{102}$$

We recalled Eqs 67 and 79 to obtain these equalities. We then proceed as for input noise to calculate

$$\langle[\theta(t) - \langle\theta(t)\rangle]^2\rangle = \int_0^t \int_0^t \mathrm{d}t'\,\mathrm{d}t'' \frac{\frac{F}{\Delta t}\displaystyle\sum_{\alpha\beta}\iint \mathrm{d}x\,\mathrm{d}y\, \sqrt{\phi[g(x)]\phi[g(y)]}\frac{\mathrm{d}\bar{g}_\alpha}{\mathrm{d}x}\frac{\mathrm{d}\bar{g}_\beta}{\mathrm{d}y}\langle\eta_\alpha(x + \theta(t'), t')\eta_\beta(y + \theta(t''), t'')\rangle}{4\tau^2\left[\displaystyle\int \mathrm{d}x\, \phi'[g(x)]\left(\frac{\mathrm{d}g}{\mathrm{d}x}\right)^2\right]^2}, \tag{103}$$

which yields the diffusion coefficient

$$D_{\text{spike}} = \frac{F\displaystyle\int \mathrm{d}x\, \phi[g(x)]\left(\frac{\mathrm{d}g}{\mathrm{d}x}\right)^2}{4\tau^2\left[\displaystyle\int \mathrm{d}x\, \phi'[g(x)]\left(\frac{\mathrm{d}g}{\mathrm{d}x}\right)^2\right]^2}. \tag{104}$$

After applying Eq 88 for a ReLU $\phi$ and setting $F = 1$ for Poisson spiking, we obtain Eq 20 of the Results section.

**Drift velocity $v_{\text{conn}}(\theta)$ due to quenched connectivity noise.** Suppose that we perturb the symmetric, translation-invariant $W$ by a small component $V$ representing deviations away from an ideal attractor architecture:

$$W_\beta(x, y) \rightarrow W_\beta(x, y) + V_{\alpha\beta}(x, y). \tag{105}$$

By Eq 33, this produces the noise term

$$\zeta_\alpha(x, t) = \sum_\beta \int \mathrm{d}y \, V_{\alpha\beta}(x, y) \phi[g_\beta(y, t)]. \tag{106}$$

In contrast to input and spiking noise, this noise is correlated across neurons and time, so it cannot be averaged away as in Eqs 89 and 90. Substituting Eq 106 into Eq 84, we obtain

$$\frac{\mathrm{d}\theta}{\mathrm{d}t} = v_{\text{drive}} + v_{\text{conn}}(\theta), \tag{107}$$

where the drift velocity is

$$v_{\text{conn}}(\theta) = -\frac{\sum_{\alpha\beta} \iint \mathrm{d}x \, \mathrm{d}y \, V_{\alpha\beta}(x, y) \phi'[g(x - \theta)] \frac{\mathrm{d}g(x - \theta)}{\mathrm{d}x} \phi[g(y - \theta)]}{2\tau \int \mathrm{d}x \, \phi'[g(x - \theta)] \left(\frac{\mathrm{d}g(x - \theta)}{\mathrm{d}x}\right)^2}. \tag{108}$$

Because $V$ is already small, we ignored $\xi$ in Eq 77 to obtain this expression. We have also made the dependence on bump position $\theta$ explicit to illustrate how it influences $v_{\text{conn}}(\theta)$. After applying Eq 88 for a ReLU $\phi$, we obtain Eq 24 of the Results section.

We now make scaling arguments for speed difference (Eq 30), speed variability (Eq 31), and escape drive $b_0$ (Eq 26). To do so, we impose a ReLU $\phi$ and return to discrete variables to be explicit:

$$v_{\text{conn};\theta} = -\frac{\sum_{\alpha\beta} \sum_{ij} V_{\alpha\beta ij} \cdot \Delta s_{i-\theta} \cdot s_{j-\theta}}{2\tau \sum_i (\Delta s_{i-\theta})^2}. \tag{109}$$

We need to understand how the numerator scales with $M$ and $N$. It is a weighted sum of $4N^2$ independent Gaussian random variables $V_{\alpha\beta ij}$ and is thus a Gaussian random variable itself. It has zero mean, but its variance is proportional to $N^2 \cdot M^2/N^2$. The $N^2$ comes from the number of terms in the sum and the $M^2/N^2$ comes from the scaling of $\mathrm{d}s/\mathrm{d}x$ (Eq 11). In combination with the scaling of the denominator, we conclude that $v_{\text{conn};\theta}$ is a Gaussian random variable with

$$\mathrm{E}[v_{\text{conn};\theta}] = 0, \qquad \mathrm{Var}[v_{\text{conn};\theta}] \propto \frac{N^2}{M^2}. \tag{110}$$

Eq 109 implies that $v_{\text{conn};\theta}$ is correlated over $\theta$. The weights for the sum over $V_{\alpha\beta ij}$ are the firing rates and their derivatives for a bump centered at $\theta$. If $\theta$ is slightly changed, almost the same entries of $V$ will be summed over with similar weights. The amount of correlation across $\theta$ is determined by the degree of overlap in weights, and therefore, by the width and number of bumps. Let's consider the effects of changing $N$ and $M$ on the covariance matrix $\mathrm{Cov}[v_{\text{conn};\theta}$,

$v_{\text{conn};\theta}$]. A larger $N$ increases the bump width and the correlation length proportionally, so values of the main diagonal decay proportionally more slowly into the off diagonals. A larger $M$ redistributes values among the diagonals by decreasing the bump width and adding more bumps, but it does not change the total amount of correlation. Thus,

$$\sum_{\theta,\theta'} \text{Cov}[v_{\text{conn};\theta}, v_{\text{conn};\theta'}] \propto N^2 \cdot \text{Var}[v_{\text{conn};\theta}]. \tag{111}$$

This allows us to evaluate

$$\text{Var}\left[\underset{\theta}{\text{mean}}\, v_{\text{conn};\theta}\right] = \text{Var}\left[\frac{1}{N}\sum_{\theta} v_{\text{conn};\theta}\right] = \frac{1}{N^2}\sum_{\theta,\theta'}\text{Cov}[v_{\text{conn};\theta}, v_{\text{conn};\theta'}] \propto \frac{N^2}{M^2}. \tag{112}$$

As a sum of zero-mean Gaussian random variables, $\text{mean}_{\theta}\, v_{\text{conn};\theta}$ is also a zero-mean Gaussian random variable. That means $|\text{mean}_{\theta}\, v_{\text{conn};\theta}|$ follows a folded normal distribution, which obeys

$$\text{E}\left[\left|\underset{\theta}{\text{mean}}\, v_{\text{conn};\theta}\right|\right] = \sqrt{\frac{2}{\pi}\text{Var}\left[\underset{\theta}{\text{mean}}\, v_{\text{conn};\theta}\right]} \propto \frac{N}{M}. \tag{113}$$

Combining this with Eqs 12 and 14 produces the scalings for speed difference in Eq 32. We now study speed variability, which involves the expression

$$\underset{\theta}{\text{std}}\, v_{\text{conn};\theta} = \sqrt{\frac{1}{N}\sum_{\theta} v^2_{\text{conn};\theta}}. \tag{114}$$

Since each $v_{\text{conn};\theta}$ is Gaussian, the sum of their squares follows a generalized chi-square distribution. Its mean is the trace of the covariance matrix $\text{Cov}[v_{\text{conn};\theta}, v_{\text{conn};\theta'}]$, which is equal to $N$ times the variance. Thus, by Eq 110,

$$\text{E}\left[\frac{1}{N}\sum_{\theta} v^2_{\text{conn};\theta}\right] = \frac{1}{N}\cdot N \cdot \text{Var}[v_{\text{conn};\theta}] \propto \frac{N^2}{M^2}. \tag{115}$$

We are interested in the square root of the random variable on the left-hand side, and we anticipate its expected value to scale as the square root of the right-hand side. We can make this argument precise. Suppose $H$ is a random variable with a probability distribution function $p(h)$ that scales with a power of the parameter $B$. We can write

$$p(h) = B^n P(B^m h) \tag{116}$$

for exponents $n$ and $m$, where the rescaled probability distribution function $P$ does not scale with $B$. Conservation of total probability implies

$$B^n \int \text{d}h\, P(B^m h) = B^n B^{-m} \int \text{d}h'\, P(h') = 1. \tag{117}$$

Thus, $m = n$. Next, suppose we know that $\text{E}[H] \propto B^o$:

$$\text{E}[H] = B^n \int \text{d}h\, h\, P(B^n h) = B^{-n} \int \text{d}h'\, h'\, P(h') \propto B^o. \tag{118}$$

Thus, $n = -o$. We can now conclude that $E[\sqrt{H}] \propto \sqrt{E[H]}$:

$$E[\sqrt{H}] = B^{-o} \int dh \sqrt{h}\, P(B^{-o}h) = B^{o/2} \int dh' \sqrt{h'}\, P(h') \propto B^{o/2}. \tag{119}$$

Applying this result to Eq 115, we obtain

$$E\left[\operatorname*{std}_{\theta} v_{\mathrm{conn};\theta}\right] = E\left[\sqrt{\frac{1}{N}\sum_{\theta} v_{\mathrm{conn};\theta}^2}\right] \propto \sqrt{E\left[\frac{1}{N}\sum_{\theta} v_{\mathrm{conn};\theta}^2\right]} \propto \frac{N}{M}. \tag{120}$$

Combining this with Eqs 12 and 14 produces the scalings for speed variability in Eq 32.

The escape drive $b_0$ involves the expression $\max_\theta |v_{\mathrm{conn};\theta}|$. Extreme value statistics for correlated random variables is generally poorly understood. We follow Ref [78] and provide a heuristic argument for its scaling. We can partition $v_{\mathrm{conn};\theta}$ across $\theta$ into groups that are largely independent from one another based on its correlation structure. As discussed above, $v_{\mathrm{conn};\theta}$ is a weighted sum of independent Gaussian random variables $V_{\alpha\beta ij}$ (Eq 109). The weights are products between the firing rates $s_{j-\theta}$ and their derivatives $\Delta s_{i-\theta}$ for a configuration centered at position $\theta$. If we choose two $\theta$'s such that their bumps do not overlap, the corresponding $v_{\mathrm{conn};\theta}$'s will sum over different $V_{\alpha\beta ij}$'s and will be independent. Thus, $\lambda/z$ roughly sets the number of independent components, where $\lambda$ is the bump distance and $z$ is the bump width. This ratio does not change with $M$ or $N$ in our networks (Fig 2F), so the maximum function does not change the scaling of $|v_{\mathrm{conn};\theta}|$:

$$\max_{\theta}|v_{\mathrm{conn};\theta}| \propto |v_{\mathrm{conn};\theta}|. \tag{121}$$

The scaling of $E[|v_{\mathrm{conn};\theta}|]$ can be determined from $\mathrm{Var}[v_{\mathrm{conn};\theta}]$ through arguments similar to those made in Eqs 116–119. Suppose we know that $\mathrm{Var}[H] \propto B^o$ and $E[H] = 0$. Then,

$$\mathrm{Var}[H] = B^n \int dh\, h^2\, P(B^n h) = B^{-2n} \int dh'\, (h')^2\, P(h') \propto B^o. \tag{122}$$

Thus, $n = -o/2$. We can now conclude that $E[|H|] \propto \sqrt{\mathrm{Var}[H]}$:

$$E[|H|] = B^{-o/2} \int dh\, |h|\, P(B^{-o/2}h) = B^{o/2} \int dh'\, |h'|\, P(h') \propto B^{o/2}. \tag{123}$$

Applying this result to Eq 121, we obtain

$$E\left[\max_{\theta}|v_{\mathrm{conn};\theta}|\right] \propto E\left[|v_{\mathrm{conn};\theta}|\right] \propto \sqrt{\mathrm{Var}[v_{\mathrm{conn};\theta}]} \propto \frac{N}{M}. \tag{124}$$

Combining this with Eqs 12, 13 and 26 produces the scalings for the escape drive $b_0$ in Eq 27.

## Simulation methods

### Dynamics and parameter values

To simulate the dynamics in Eq 33, we discretize the network by replacing neural position $x$ with index $i$ and propagate forward in time with the simple Euler method:

$$\tau \frac{g_{\alpha i}(t + \Delta t) - g_{\alpha i}(t)}{\Delta t} + g_{\alpha i}(t) = \sum_{\beta j} W_{\beta i j} s_{\beta j}(t) + A \pm_{\alpha} \gamma b(t) + \zeta_{\alpha i}(t). \tag{125}$$

We use $\tau = 10$ ms. We use $\Delta t = 0.5$ ms and $A = 1$ for all simulations except those with spiking neurons. In the latter case, we use finer timesteps $\Delta t = 0.1$ ms and set $A = 0.1$ ms$^{-1}$. Synaptic inputs $g$ and resting inputs $A$ can be dimensionless for rate-based simulations, but they must have units of rate for spiking simulations. We use $\gamma = 0.1$ for rate-based simulations and $\gamma = 0.01$ ms$^{-1}$ for spiking simulations. In all cases, we run the simulation for 1000 timesteps before recording any data to form the bumps. To achieve the relationship in Eq 13 for circular mapping, we rescale $\gamma$ with network size $N$ and bump number $M$:

$$\gamma \rightarrow \gamma \cdot \frac{N}{600} \cdot \frac{3}{M}. \tag{126}$$

The connectivity $W$ takes the form in Eq 38. Unless otherwise specified, we use shift $\xi = 2$. To produce $M$ bumps in a network of size $N$, we turn to Eq 47 and set $l = 0.44N/M$. Note that an alternative to Eqs 13 and 126 would be to rescale $\xi$ proportionally with $l \propto N/M$ under circular mapping, since bump velocity is also proportional to $\xi$ (Fig 3C). We use $w = 8M/N \approx 3.5/l$. For the case of $2l > N/2$, which corresponds to a one-bump network, the tails of the cosine function extend beyond the network size. Instead of truncating them, we wrap them around the ring:

$$W(x) \rightarrow W(x) + W(x - N) + W(x + N). \tag{127}$$

This procedure, along with the scaling of $w$ with $N$ and $M$, accomplishes Eq 7 and keeps the total connectivity strength per neuron $\sum_i W_i$ constant across all $N$ and $M$, where $W_i$ is the discrete form of $W(x)$.

To generate the Poisson-like spike counts $c_{\alpha i}(t)$ in Eq 98, we rescale Poisson random variables:

$$c_{\alpha i}(t) = F \cdot C_{\alpha i}(t), \qquad C_{\alpha i}(t) \sim \text{Pois}[\phi[g_{\alpha i}(t)]\Delta t/F]. \tag{128}$$

These counts will be multiples of the Fano factor $F$. To produce a $c_{\alpha i}(t)$ whose domain is the natural numbers, one can follow Ref [18], which takes multiple samples of $C_{\alpha i}(t)$ during each timestep.

To obtain theoretical values in Figs 3, 5, 7 and 8, we need to substitute the baseline inputs $g_i$ into the appropriate equations. We use noiseless and driveless simulations to generate $g_i$ instead of using Eq 4.

### Bump position

We track the position $\theta$ of each bump using the firing rate summed across both populations $S_i(t) = \sum_{\alpha} \phi[g_{\alpha i}(t)]$. We first compute the circular center of mass of $S_i(t)$ with periodicity $N/M$:

$$\theta_0 = \frac{N}{2\pi M} \text{atan2}\left[\sum_i S_i(t) \sin(2\pi i M/N), \; \sum_i S_i(t) \cos(2\pi i M/N)\right]. \tag{129}$$

atan2 is the two-argument arctangent, and we choose its branch cut such that its range is [0, $2\pi$). Thus, $\theta_0$ lies between 0 and $N/M$ and represents the bump position averaged periodically across bumps. To track the position of each bump independently, we then partition the network into segments of length $\lfloor N/M \rfloor$. If $N/M$ is not an integer, we skip one neuron between some segments to have them distributed as evenly as possible throughout the network. We perform a circular shift of $S_i(t)$ such that network position $\theta_0$ is shifted to the middle of the first segment $N/2M$, after rounding both quantities to integers. The purpose of this process is to approximately center each bump within a segment so that $S_i(t)$ drops to 0 before reaching segment boundaries. We then calculate the center of mass of $S_i(t)$ within each segment. After reversing the circular shift, these centers of masses are taken to be the bump positions.

## Path integration velocity and diffusion

To obtain our results in Figs 3 and 5, we run each simulation for $T = 5$ s. To extract the bump velocity $v$ produced by a constant drive $b$, we calculate the mean displacement $\Theta$ as a function of time offset $u$:

$$\Theta(u) = \frac{\Delta t}{T - u} \sum_t [\theta(t + u) - \theta(t)]. \tag{130}$$

$\theta$ is the bump position. This equation averages over the fiducial starting time $t$, which ranges from 0 to $T - u - \Delta t$ in increments of $\Delta t$. We vary $u$ between 0 and $T/2$ in increments of $\Delta t$; the maximum is $T/2$ to ensure enough $t$'s for accurate averaging. We then fit $\Theta(u)$ to a line through the origin to obtain the velocity:

$$\Theta(u) \approx vu. \tag{131}$$

We calculate the diffusion coefficient $D$ based on an ensemble of replicate simulations. In this section, angle brackets will indicate averaging over this ensemble. Following the definition of $D$ in Eq 92, we calculate each bump's position relative to the mean motion of the ensemble:

$$\omega(t) = \theta(t) - \langle \theta(t) \rangle. \tag{132}$$

We compute squared displacements and then average over fiducial starting times to obtain a mean squared displacement for each bump as a function of time offset $u$:

$$\Omega(u) = \frac{\Delta t}{T - u} \sum_t [\omega(t + u) - \omega(t)]^2. \tag{133}$$

$t$ and $u$ span the same time ranges as they did for $\Theta$. We average $\Omega(u)$ over the ensemble and fit it to a line through the origin to obtain the diffusion coefficient:

$$\langle \Omega(u) \rangle \approx 2Du. \tag{134}$$

For simulations with $M$ bumps, we arbitrarily assign identity numbers 1, . . ., $M$ to bumps in each simulation. We perform ensemble averaging over bumps with the same identity numbers; that is, we only average over one bump per simulation. This way, we obtain separate values for each bump in Fig 3E–3H; nevertheless, these values lie on top of each other. In Fig 3B and 3C, each point represents $v$ averaged across bumps. To calculate the mean velocity $\langle v \rangle$ in Fig 3E and 3F, we fit $\langle \Theta(u) \rangle$ to a line through the origin. To estimate standard deviations for Figs 3E–3H and 5, we create 48 bootstrapped ensembles, each of which contains 48 replicate simulations sampled with replacement from the original ensemble. We calculate $\langle v \rangle$ or $D$ for

each bootstrapped ensemble and record the resulting standard deviation. In Fig 5, each point represents $D$ and its estimated standard deviation averaged across bumps.

## Trapping and position-dependent velocity

For simulations with connectivity noise, we determine the escape drive $b_0$ (Fig 7), the smallest drive that allows the bumps to travel through the entire network, by a binary search over $b$. We perform 8 rounds of search between the limits 0 and 1.28 and another 8 rounds between 0 and $-1.28$ to obtain $b_0$ within an accuracy of 0.01. In each round, we run a simulation with the test $b$ and see whether the bumps travel through the network or get trapped. Traveling through the network means that every position (rounded to the nearest integer) has been visited by a bump, and trapping means that the motion of at least one bump slows below a threshold for a length of time.

To obtain the position-dependent bump velocity $v(\theta)$ produced by connectivity noise when $|b| > b_0$, we run a simulation until the bumps have traveled through the network. At each time-step, we record the positions of the bumps (binned to the nearest integer) and their instantaneous velocities with respect to the previous timestep. We smooth the velocities in time with a Gaussian kernel of width 10 ms, which is the neural time constant $\tau$. We compute the mean and standard deviation of these smoothed velocities for each position bin.

## Mutual information

For simulations with input noise, we explore the mutual information between encoded coordinate and single-neuron activity (Fig 6). To do so, we must generate data from which we can compute $p(s|u)$ in Eq 22, for coordinate $u \in \mathcal{U}$ and activity $s \in \mathcal{S}$. We have chosen one set of conditions for performing this analysis, which we detail below.

We first choose to represent either a linear or circular coordinate, which we take to be position or orientation, respectively. We then choose to represent a narrow or wide coordinate range $u_{\max}$, which is 20 cm or 200 cm for position and 36˚ or 360˚ for orientation. We divide the range into 20 equally spaced coordinates such that $\mathcal{U} = \{u_{\max}/20, \dots, u_{\max}\}$. We convert these coordinates to network positions according to the mappings in Fig 4. For each coordinate value $u$, we initialize 96 replicate simulations at the corresponding network position by applying additional synaptic input to the desired bump positions during bump formation. We run the simulations for 5 s, record the final firing rates, and bin them using 6 equally spaced bins from 0 to the 99th percentile across all neurons. All rates above the 99th percentile are also added to the 6th bin. These bins define the discrete $\mathcal{S}$, and normalizing the bin counts produces $p(s|u)$. We marginalize over $u$ to obtain $p(s)$, and $p(u)$ is uniform. We can then use Eq 22 to compute the mutual information.

The 4 local cues in Fig 6F–6H correspond to 4 activity states $\mathcal{S}_{\text{cue}}$ separate from the 6 activity bins of the CAN neurons, $\mathcal{S}_{\text{neuron}}$. The joint sample space of a single neuron with cues is thus $\mathcal{S} = \mathcal{S}_{\text{neuron}} \times \mathcal{S}_{\text{cue}}$ with $6 \times 4 = 24$ total states. We bin neural activity across these more numerous states, using the coordinate value $u$ to determine the cue state value, to again compute $p(s|u)$ and then the mutual information.

We choose to compute mutual information with single-neuron activities binned into 6 discrete states due to computational tractability. A better indication of encoding quality for the entire network would involve using the joint activity of multiple neurons. However, assuming the same binning process, that would involve estimating probability distributions over $6^n$ states for $n$ neurons, which would require exponentially more replicate simulations per coordinate value than the 96 we use. Alternatively, one could reduce the dimensionality of the network activity by projecting it onto various attractor configurations, as done by Ref [79].

## Supporting information

**S1 Text. Contains text on different model parameters, additional information results, and splitting networks, as well as Figs A, B, and C.**
(PDF)

## Acknowledgments

We are grateful to Steven Lee for sharing his code and to John Widloski for a careful reading of this manuscript.

## Author Contributions

**Conceptualization:** Louis Kang.

**Formal analysis:** Louis Kang.

**Funding acquisition:** Louis Kang.

**Investigation:** Raymond Wang, Louis Kang.

**Software:** Raymond Wang, Louis Kang.

**Supervision:** Louis Kang.

**Visualization:** Raymond Wang, Louis Kang.

**Writing – original draft:** Raymond Wang, Louis Kang.

**Writing – review & editing:** Raymond Wang, Louis Kang.

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
