## [Decision Letter · Decision Letter 0]

5 May 2022

Dear Dr. Kang,

Thank you very much for submitting your manuscript "Multiple bumps can enhance robustness to noise in continuous attractor networks" for consideration at PLOS Computational Biology.

As with all papers reviewed by the journal, your manuscript was reviewed by members of the editorial board and by several independent reviewers. In light of the reviews (below this email), we would like to invite the resubmission of a significantly-revised version that takes into account the reviewers' comments.

We cannot make any decision about publication until we have seen the revised manuscript and your response to the reviewers' comments. Your revised manuscript is also likely to be sent to reviewers for further evaluation.

Sincerely,

Xuexin Wei

Associate Editor

PLOS Computational Biology

Daniele Marinazzo

Deputy Editor

PLOS Computational Biology

Reviewer's Responses to Questions

**Comments to the Authors:**

Reviewer #1: The authors analyze and simulate the dynamical evolution of bump solutions in perturbed versions of continuous attractor networks. Their analysis relies primarily on weak noise or weak connectivity perturbation assumptions so that the stochastic motion of bumps can be approximated by linearizing around stationary bump or pattern solutions. As the authors make clear, these approaches and findings are actually fairly standard at this point and already developed and used by a number of authors to analyze stochastic bump dynamics in neural network models that are similar but maybe not quite quantitatively parameterized the same. The new finding proffered by this paper is simply that for the particular model parameterization increasing the number of bumps decreases bump diffusion for "linear" networks and increases bump diffusion for "circular" networks. Linear networks reside on a non-periodic line and circular networks are periodic.

My primary issue with the work is that the coupling strength scaling (Eq. 13) the authors choose seems to be doing a lot of the work in shaping these relations, but this is not well justified. Moreover, I do not see how general this result is or if you've simply chosen constituent functions of the model (\\phi and W) that are non-generic and so produce a behavior that would not likely persist in other networks with different transfer or weight functions. In fact, in a recent review by Stein, Barbosa, & Compte (2021; Curr Op Neuro), this issue is alluded to, in that qualitative trends in bump diffusion increasing/decreasing can be model function choice dependent. I don't think the authors are really hiding anything, but I would like to see how well these results hold up across a broader range of model function choice. Otherwise, I'd say given the technical nature of the paper, it is presented thoroughly and clearly. See below for more detailed comments.

Comments:

1. You seem to suggest that networks with M bumps essentially code information the same way regardless of bump number. However, a single neuron will see activity rise and fall M times in a "rotation period" and so the same velocity integration process can represent an interval of 1/M the length. This is made fairly clear in past papers of grid cell models that the authors cite. It is unclear how bump identity could be known by a single neuron (in Fig. 1). It seems like you would shorten the period. What is driving the rotation of the bumps? Is it now that you have to pack neurons of a particular tuning N times as tightly?

2. The intro is wrong in asserting no one else has examined the effect of bump number of noise-induced WM degradation. Many past papers have:

Edin et al. Mechanism for top-down control of working memory capacity (2009) PNAS

Wei et al. From distributed resources to limited slots in multiple-item working memory: a spiking network model with normalization (2012) J Neurosci

Almeida et al. Neural circuit basis of visuo-spatial working memory precision: a computational and behavioral study (2015) J Neurophysiol

Krishnan et al. Synaptic efficacy shapes resource limitations in working memory (2018) J Comput Neurosci

Bouchacourt & Buschman. A flexible model of working memory (2019) Neuron

3. Per my point above, what justifies rescaling connectivity as in Eq 7? A lot of results in the paper hinge on this, but it is not well justified.

4. Provide justification/citation for the reduction from Eq 1 to Eq 3. Does this depend on the ReLu transfer function assumption? You do this in the Theoretical model section too, but don’t really prove the reduction is valid.

5. Line 119: Doesn’t the period of the network impact the spacing more than a small difference? Seems like this would be the case especially if L/\\lambda equals an integer plus a half.

6. Why does keeping the bump shape the same make for a fair comparison? Neural architecture would seem more likely to be the constrained quantity. You should provide more evidence that there is some straightforward mechanism for the amplitude of neural connectivity to be varied so that bump are of the same size regardless of their number. This seems like it would be a complicated feedback control mechanism.

7. You introduce the notion of linear vs circular coordinates, but you are not very clear about what boundary conditions you use in the linear coordinate case. Are they free boundaries? In general you are on clear about what weight function W is used in any of the plots. Is this somewhere buried in the methods at the end?

8. Line 355: You say most of your results apply to a general nonlinear transfer function. Where do you make it clear that this is true? All your plots are for the ReLU, so I’m not sure where you’re showing results for other nonlinearities. Do Eq. 8, 10, 12 all hold for general \\phi or specifically the ReLU? Make this more clear.

9. Your analysis assumes periodic solutions g that are sinusoidal, but shouldn’t there also be solutions that might just be single localized bumps? Did you explore this possibility? Is it clear that choosing W as in Eq. 37 rules out such solutions? If so, it seems to me that the W you chose is nongeneric, since typically in continuum bump attractor networks we should find single bumps with quiescence around.

Reviewer #2: The paper by Wang and Kang addresses an important problem in theoretical neuroscience. Continuous attractors are an important conceptual model in neuroscience but it is know that noise can cause drift them. This will hamper both the retention of information in continuous attractor models of working memory and the integration of input in continuous attractor models of path integration.

The main contribution of the paper is that noise induced drift can be minimized by using connectivity patterns that yield more bumps. This is shown both by solid analytical calculations and computer simulations.

In general this is a good paper, but I think it suffers from two major and one minor issues as I describe below.

Major Issues:

1. Although I largely agree with the author's finding that drift due to fast noise can be reduced by having more bumps, I do not find that this necessarily confers, or that the authors show that it confers, a benefit when looked at from the perspective of the presumed function of continuous attractors. Having more bumps in a network of N neurons, in addition to reducing noise, reduces the resolution by which the external covariate can be encoded. In other words, what the authors do not address is how much "information" is lost or maintained as a result of drift with different number of bumps. Consider using a continuous attractor of N neurons for keeping an angle in working memory. The original angle, encoded by the original position of the bump(s), and the angle encoded by the position of the bump(s) after a given time T may be closer to each other when there are more bumps in the network. However, with one bump, the 360 degrees can be represented in steps of 360/N degrees, while with two bumps in steps of 360*2/N. I think the authors should perhaps use mutual information to study this tradeoff as has been used e.g. in Roudi and Treves PLoS Comp Biol 2008.

2. The only focus on the effect of fast noise on the drift. If not more important, an equally important problem, is that of the noise induced by quenched noise, e.g. when the connectivity is not a perfect connectivity in Fig2 b but with a temporally fixed noise added to it, e.g. see Eq. 2 Itskov et al 2011. I think the authors should discuss the interaction between multi- bumps solutions and the influence of quenched noise in more detail perhaps also discussing the recent result in https://arxiv.org/pdf/2109.03879.pdf

Minor:

There is also a minor issue regarding the way the authors discuss CANs. The authors seem to confuse evidence that is compatible with CANs with evidence that shows the existence of CANs in the brain. As far as I can say, none of the papers cited by the authors show that CANs are actually implemented in the brain or are actually used by the brain. The cited evidence are either theoretical models or experimental data that are consistent or supportive of CANs. I suggest the authors be more careful (e.g. "The brain uses path-integrating CANs" can be changed to "Path integrating CANs have been proposed as a mechanism ...").

**Have the authors made all data and (if applicable) computational code underlying the findings in their manuscript fully available?**

Reviewer #1: **No: **The response to this question the authors give is simply the link https://louiskang.group/repo but this does not include a folder for code for the current paper.

Reviewer #2: Yes

PLOS authors have the option to publish the peer review history of their article (what does this mean?). If published, this will include your full peer review and any attached files.

Reviewer #1: No

Reviewer #2: No
---

## [Decision Letter · Decision Letter 1]

30 Jul 2022

Dear Dr. Kang,

Thank you very much for submitting your manuscript "Multiple bumps can enhance robustness to noise in continuous attractor networks" for consideration at PLOS Computational Biology. As with all papers reviewed by the journal, your manuscript was reviewed by members of the editorial board and by several independent reviewers. The reviewers appreciated the attention to an important topic. Based on the reviews, we are likely to accept this manuscript for publication, providing that you modify the manuscript according to the review recommendations.

Sincerely,

Xuexin Wei

Associate Editor

PLOS Computational Biology

Daniele Marinazzo

Deputy Editor

PLOS Computational Biology

[LINK]

Reviewer's Responses to Questions

**Comments to the Authors:**

Reviewer #1: Thanks to the authors for addressing my concerns. I now support publication of the article in PLoS Computational Biology.

Reviewer #3: The continuous attractor model is believed to represent the dynamics and structure of certain populations of neurons in the brain – such as head direction cells and grid cells, thought to encode the direction the animal is facing and its location. The networks are usually given a distinct topology and a predefined connectivity, and the activity of each neuron is given by the time-dependent synaptic input. Depending on the connectivity, one or more activity packets or “bumps” arise on the network. These may be smoothly translated along the network and the position of the packet(s) usually depict the current state of some external variable – e.g. position in space or head direction. It is still an open question whether the brain follows continuous attractor dynamics and if so, whether the networks hold one or more bumps.

Wang and Kang give a convincing exposé of how, under certain circumstances, there may be an advantage in a continuous attractor network (CAN) having multiple bumps. The authors provide a thorough analysis on the effect of three variants of noise - synaptic input (1), spiking noise (2) and perturbations in connectivity (3) - on a 1D CAN model with ring topology under varying bump numbers and network size. The measure of robustness is given through bump diffusion coefficients and mutual information for noise 1 and 2 and by bump “escape drive” and bump velocity irregularities for 3. Through mathematical derivations the noise dependence on number of bumps and neurons is clearly shown, and the findings are supported by simulations. The authors conclude that, for CANs representing a linear variable (e.g. location on a linear track), the robustness to noise is increased when having more bumps but decreased with a large number of neurons. For CANs representing a circular variable (e.g. head direction) there is no effect of adding more bumps, but increasing the number of neurons makes the network more robust. The figures are polished and easy to understand, and the text well-written. The code is neatly organized, easy to implement and runs quickly.

The revision has improved the paper by adding mutual information as a second measure of robustness, clarifying certain statements, including references to working memory CAN literature and adding an analysis of a different non-linear activation function as well as an analysis using unscaled connectivity. Notably, the latter showed that fewer bumps had greater robustness to noise for the circular mapping.

The analysis is elaborate and the theory sound. However, such analysis is dependent on the assumptions made in the model. The authors explore a few different conditions – linear vs circular mapping, ReLU and sigmoidal activation function, scaled vs unscaled connectivity weights and three types of noise. The specific results rely heavily on the mapping between internal and external variables and possibly other choices made throughout – e.g. the topology and dimension of the network. The plausibility for the assumptions is only briefly discussed and the intuition behind the assumptions and their consequences should be clarified to the reader. Thus, to me, the impact seems to lie rather in the framework than in the results and a main concern is the extent to which the title and main text focuses on the increased robustness of multi-bump networks. Clarifying and softening the claims may help to not give the impression that the increased robustness is a general result for all CANs.

In particular, explaining more explicitly how the choice of mapping relates to the results and why multiple bumps and a low number of neurons is beneficial for a linear mapping but not a circular one would make it easier to understand the non-triviality of the findings and thus the impact of the paper. E.g. doubling the number of bumps doubles the number of neurons encoding each (fixed) location under both linear and circular mapping. I.e. going from one to two bumps is similar to splitting the population into two networks, each encoding half the range of the original network but with equal intervals between each encoded position in the linear case, and the full range for circular mapping, but with twice the interval between each angle. Could the results thus depend on the number of neurons encoding each (fixed) interval, the size of the interval, the total range of coordinates encoded by the network or perhaps just the "spread" of noise across bumps? How do the results compare when splitting the networks - is the combined readout of multiple networks more or less robust than the readout of one network with multiple bumps and which is more biologically plausible? Showing more intuitively how the results relate to the assumptions would make the exposition clearer and help understand what is really making a difference.

Lines 130-131 say the scaling of W removes any influence of bump shape. However, the change of firing rate over network position (i.e. the slope of the bump ds/dx ~ M/N = 1/\\lambda), seems to be an important factor for robustness (e.g. eq. 10). It would seem the results could be explained by saying that adding more bumps (in the linear case) decreases the bump distance, increasing ds/dx (hence changing the bump shape) and thus increasing robustness (while under circular mapping ds/dx is unchanged as the network distance is scaled)? In 10A, we see how the change in w changes the shape of the bump (and ds/dx) as seen from eq. 58 and 60, which should also be reflected in the results. A clearer derivation/analysis of this effect would be enlightening.

Similarly, it could be explicitly shown in Fig. 6 how the temporal/spatial shape of the bumps changes when introducing more bumps and how this leads to a different spatial tuning curve per neuron and thus a higher mutual information score for the linear case. In this regard, I am also missing mutual information scores in Fig. 10.

In computing mutual information, I am wondering if it would be fairer to look at the whole coordinate range of the network? Looking at different number of bumps is similar to studying neurons of grid cell modules of different scales and I would expect the descriptive power of each neuron to be the same but at different spatial scales. Visual cues (or multiple modules) may be involved in confining the possible states, but would the results then be obvious in that grid modules with small spacing gives a more precise description of location (in a short interval) than those with large spacing?

It would also be interesting to see how robust the readout of the combined network of grid cell modules (of different spacing) would be with a varying number of bumps and how they could interact to resolve the ambiguity.

**Have the authors made all data and (if applicable) computational code underlying the findings in their manuscript fully available?**

Reviewer #1: **No: **I don't see a code repo link.

Reviewer #3: Yes

PLOS authors have the option to publish the peer review history of their article (what does this mean?). If published, this will include your full peer review and any attached files.

Reviewer #1: No

Reviewer #3: **Yes: **Erik Hermansen

Figure Files:

Data Requirements:

Reproducibility:

References:

---

## [Decision Letter · Decision Letter 2]

6 Sep 2022

Dear Dr. Kang,

We are pleased to inform you that your manuscript 'Multiple bumps can enhance robustness to noise in continuous attractor networks' has been provisionally accepted for publication in PLOS Computational Biology.

Best regards,

Xuexin Wei

Academic Editor

PLOS Computational Biology

Daniele Marinazzo

Section Editor

PLOS Computational Biology

Reviewer's Responses to Questions

**Comments to the Authors:**

Reviewer #3: I thank the authors for addressing my concerns, their findings are now more intuitively explained through revised/added sentences and with the discussion of Fisher information. The augmentation of analysis of multiple networks and added mutual information results showcase the ambiguity of the advantage of multiple bumps (Split networks with fewer bumps can have similar noise resilience and the mutual information results describe rather the range/resolution (and hence possibly function) of the network than noise robustness), making the article more well-balanced.

The article serves as a nice addition to the CAN literature and I support its publication in PLoS Computational Biology.

**Have the authors made all data and (if applicable) computational code underlying the findings in their manuscript fully available?**

Reviewer #3: Yes

PLOS authors have the option to publish the peer review history of their article (what does this mean?). If published, this will include your full peer review and any attached files.

Reviewer #3: **Yes: **Erik Hermansen

---

## [Editor Report · Acceptance letter]

28 Sep 2022

PCOMPBIOL-D-22-00269R2 

Multiple bumps can enhance robustness to noise in continuous attractor networks

Dear Dr Kang,

I am pleased to inform you that your manuscript has been formally accepted for publication in PLOS Computational Biology. Your manuscript is now with our production department and you will be notified of the publication date in due course.

With kind regards,

Zsofia Freund
